# Levels of complement factor H-related 4 protein do not influence susceptibility to age-related macular degeneration or its course of progression

M. A. Zouache ⬤[1,2] ✉, B. T. Richards[1,2], C. M. Pappas[1], R. A. Anstadt[1], J. Liu[1], T. Corsetti[1], S. Matthews[1], N. A. Seager[1], S. Schmitz-Valckenberg ⬤[1], M. Fleckenstein[1], W. C. Hubbard[1], J. Thomas[1], J. L. Hageman[1], B. L. Williams[1] & G. S. Hageman[1] ✉

Dysregulation of the alternative pathway (AP) of the complement system is a significant contributor to age-related macular degeneration (AMD), a primary cause of irreversible vision loss worldwide. Here, we assess the contribution of the liver-produced complement factor H-related 4 protein (FHR-4) to AMD initiation and course of progression. We show that FHR-4 variation in plasma and at the primary location of AMD-associated pathology, the retinal pigment epithelium/Bruch's membrane/choroid interface, is entirely explained by three independent quantitative trait loci (QTL). Using two distinct cohorts composed of a combined 14,965 controls and 20,741 cases, we ascertain that independent QTLs for FHR-4 are distinct from variants causally associated with AMD, and that FHR-4 variation is not independently associated with disease. Additionally, FHR-4 does not appear to influence AMD progression course among patients with disease driven predominantly by AP dysregulation. Modulation of FHR-4 is therefore unlikely to be an effective therapeutic strategy for AMD.

Age-related macular degeneration (AMD) affects close to 200 million individuals worldwide[1] and accounts for approximately 15–20% of all causes of irreversible vision loss among individuals over 50 in Europe and North America[2]. The critical site for initiation and progression of AMD lies at the outer blood-retinal barrier, which mainly consists of the retinal pigment epithelium (RPE), underlying Bruch's membrane and choriocapillaris, the capillary bed of the choroid sustaining both RPE and photoreceptors[3]. It is within this unique tissue complex that the clinical manifestations and histological hallmarks of AMD such as pigmentary abnormalities, drusen, and/or pigment epithelium detachments initiate and evolve[4]. Patients presenting these phenotypic features are at elevated risk of progressing to more advanced forms of the disease, characterized by gradual atrophy of regions of the retina (complete RPE and outer retinal atrophy, cRORA) and/or abnormal growth of structurally impaired choroidal and/or retinal blood vessels (neovascular AMD)[5]. The acute and often permanent vision loss that these two late forms of AMD cause significantly degrades the quality of life of those affected[6,7]. Interventional therapies have so far only been successfully marketed for neovascular AMD; although, they are often palliative and suffer from inconsistent clinical outcomes[8–13].

Genetic factors explain between 46% and 71% of the variation in overall severity of AMD[14] and play a significant role in the initiation,

[1]Sharon Eccles Steele Center for Translational Medicine, John A. Moran Eye Center, Department of Ophthalmology and Visual Sciences, University of Utah, Salt Lake City, UT, USA. [2]These authors contributed equally: M. A. Zouache, B. T. Richards. ✉e-mail: moussa.zouache@hsc.utah.edu; gregory.hageman@hsc.utah.edu

progression, and phenotypic presentation of the disease[15,16]. The most common genetic contributors to AMD among individuals with European ancestry are variants associated with a cluster of genes involved in regulation of the alternative complement pathway (AP) on chromosome 1q32 (Chr1 AMD locus) and variants associated with two neighboring genes on chromosome 10q26 (Chr10 AMD locus), age-related maculopathy susceptibility 2 (*ARMS2*) and high-temperature requirement factor A1 (*HTRA1*)[17,18]. Other genetic factors, including the thirty-two additional loci associated with AMD through genome-wide association study (GWAS)[19,20], have a comparatively smaller contribution to the variability in disease explained by genetic effects[19,21]. An increasing body of evidence indicates that the initial AMD-associated pathological events driven by Chr1 and Chr10 risk variants are distinct[15,16,22]. Therapies targeting complement dysregulation in subjects with variants associated with elevated risk at the Chr1 AMD locus represent some of the best opportunities to prevent or slow the progression of AMD. Exhaustive diplotype analyses suggest that complement-based therapeutics may also provide efficacy among patients with AMD driven by risk variants at the Chr10 AMD locus[23].

The chromosome 1q32 locus encompasses several genes encoding regulators of complement activation including complement factor H (*CFH*) and the highly homologous complement factor H-related gene family (*CFHR1, CFHR2, CFHR3, CFHR4* and *CFHR5*)[24–30]. Both risk- and protection-conferring genetic polymorphisms and haplotypes have been identified within the Chr1 AMD locus[23]. The most studied risk-associated *CFH* polymorphism, tagged by rs1061170, encodes a tyrosine to histidine substitution at amino acid position 402 of factor H protein (FH) and its alternative splice variant factor H-like protein 1 (FHL-1), both negative regulators of the AP. The FH/FHL-1 Y402H substitution alters binding affinity to several host ligands that can eventually lead to complement overactivation and pathological consequences at the RPE/Bruch's membrane/choroid interface[31–34]. Two independent protective genetic polymorphisms or haplotypes have also been uncovered within the Chr1 AMD locus. The first form of genetic protection is conferred by a common polymorphism, tagged by rs800292, that encodes a valine to isoleucine substitution at amino acid position 62 of FH/FHL-1. The I62 allotype confers increased complement co-factor and C3 convertase decay acceleration activities that result in reduced AP activation[35,36]. The second form of AMD protection arises from a deletion of both the *CFHR3* and *CFHR1* genes (*CFHR3/1* deletion), two genes that are expressed by liver, but not ocular tissue[27,30,37]. Evidence suggests that FHR-1 and FHR-3 can compete with FH and FHL-1 for binding self-surfaces and key ligands including malondialdehyde adducts and C3b, leading to AP overactivation[38–40]. Altogether, both forms of genetic protection, working via independent mechanisms, result in more optimal FH/FHL-1 AP negative regulation at the RPE/Bruch's membrane/choroid interface that delay or prevent AMD initiation and/or progression.

Several studies[41–44] have suggested, analogous to FHR-1 and FHR-3, that complement factor H-related 4 protein (FHR-4) can compete with FH/FHL-1 to alter AP activation and, therefore, that modulation of FHR-4 may be an attractive AMD therapeutic strategy. Two *CFHR4* mRNA splice variants that encode FHR-4A (~86 kDa) and FHR-4B (~43 kDa) proteins have been identified[45]. Recent ELISA studies, using isoform-specific capture/detector antibodies, have convincingly demonstrated that FHR-4A is the predominant protein in serum and that FHR-4B is below detection limits or not present in serum[46]. In this study we focus on total FHR-4 protein measured in plasma and ocular compartments and seek to elucidate its contribution to AMD susceptibility and course of progression. One of the first findings implicating FHR-4 in the pathophysiology of AMD came from a GWAS that identified a correlation between the *CFHR4* intron 1 variant rs6684931, systemic complement activation levels (serum C3d/C3 ratio) and increased risk for AMD[41]. However, because this variant is in high linkage disequilibrium (LD) with several common single nucleotide

polymorphisms (SNP) within the Chr1 AMD locus ($r^2 > 0.8$), its effect on complement activation could not be conclusively linked solely to *CFHR4*. Even though *CFHR4* mRNA is not expressed in ocular tissue, FHR-4 protein has been detected in Bruch's membrane and between capillaries of the choriocapillaris[43]. It was also shown to be present in drusen and to colocalize with complement activation products at the RPE/Bruch's membrane/choroid interface in AMD eyes[43]. The functional significance of these findings and their implication in the pathogenesis of AMD is not fully understood. Similar to the other four FHR proteins, FHR-4 has a high level of amino acid homology with SCR domains of FH but lacks critical regions responsible for co-factor and decay acceleration activity (see Fig. 1a). The prevailing paradigm is that FHR-4 reduces the ability of FH/FHL-1 to regulate AP activation[47]. It may do so by competing with FH/FHL-1 for binding to C3b[43], analogous to the FHR-1 and FHR-3 competition model[40]. Binding of FHR-4 to C3b was shown to facilitate the assembly of a functionally active C3b convertase (C3bBbP) that is less susceptible to FH-mediated decay[43,48]. In addition, FHR-4 may activate the classical pathway through its binding of C-reactive protein[49]. Altogether, these findings suggest that reducing levels of FHR-4 at the RPE/Bruch's membrane/choroid interface may lead to enhanced AP regulation by FH/FHL-1 and delay or prevent AMD initiation and/or progression. This hypothesis is partially supported by the observations that individuals carrying a deletion of *CFHR1* and *CFHR4* (*CFHR1/4* deletion), which is rare among individuals with European ancestry, are generally protected against AMD[50,51], and that circulating levels of FHR-4 are lower in healthy individuals as compared to patients with advanced AMD[43].

Here, we show that variations in FHR-4 levels in plasma and at the primary location of AMD-associated pathology, the RPE/Bruch's membrane/choroid interface, are entirely explained by three independent quantitative trait loci (QTLs). Using two distinct case/control cohorts composed of a combined 14,965 controls and 20,741 cases, we demonstrate that these *CFHR4* QTLs are distinct from variants causally associated with AMD, and that FHR-4 variation is not independently associated with disease. Survival analyses performed among patients with AMD driven predominantly by AP dysregulation additionally reveal that FHR-4 levels do not influence disease progression course either. This study demonstrates that *CFHR4* does not play an independent role in AMD disease initiation or progression, and that modulation of FHR-4 is unlikely to be an effective therapeutic strategy for this blinding disease.

## Results
### Refinement of genetic associations within the Chr1 AMD locus
We first refined genetic associations within the *CFH-CFHR5* extended region to identify causal AMD variants and implement meaningful stratifications of human subjects and donors. To do so we performed single-variant, haplotype, and diplotype association analyses using 1587 controls and 3200 cases recruited between 1999 and 2019 at the University of Utah, Salt Lake City, Utah, United States or University of Iowa, Iowa City, Iowa, United States (Utah & Iowa cohort, Supplementary Table 1)[23]; and 17,541 cases and 13,378 controls recruited by the International AMD Genomic Consortium (IAMDGC) as part of a GWAS[19] (IAMDGC Cohort, Supplementary Table S1). Eight credible sets of variants have been independently associated with AMD within the extended *CFH-CFHR5* region through GWAS[19]. Out of the eight index SNPs (IAMDGC Locus 1.1–1.8) describing these sets, four (IAMDGC Loci 1.3, 1.4, 1.7, and 1.8) are rare (frequency <1%) and were therefore excluded from further analyses. IAMDGC Locus 1.2 is in perfect linkage disequilibrium with the risk-associated rs1061170 (*CFH* Y402H, OR$_{IAMDGC}$: 2.18 [2.10; 2.26], $p = 1.1 \times 10^{-422}$); see Supplementary Table 2. The non-coding IAMDGC Locus 1.1 (OR$_{IAMDGC}$: 0.44 [0.42; 0.46], $p = 4.5 \times 10^{-415}$), tagged by rs10922109 and rs1410996 ($r^2 = 0.9919$, $D' = 0.9959$) is generally used as a proxy for the combination of haplotypes containing the rs800292 variant (*CFH* I62V, OR$_{IAMDGC}$: 0.51 [0.49;

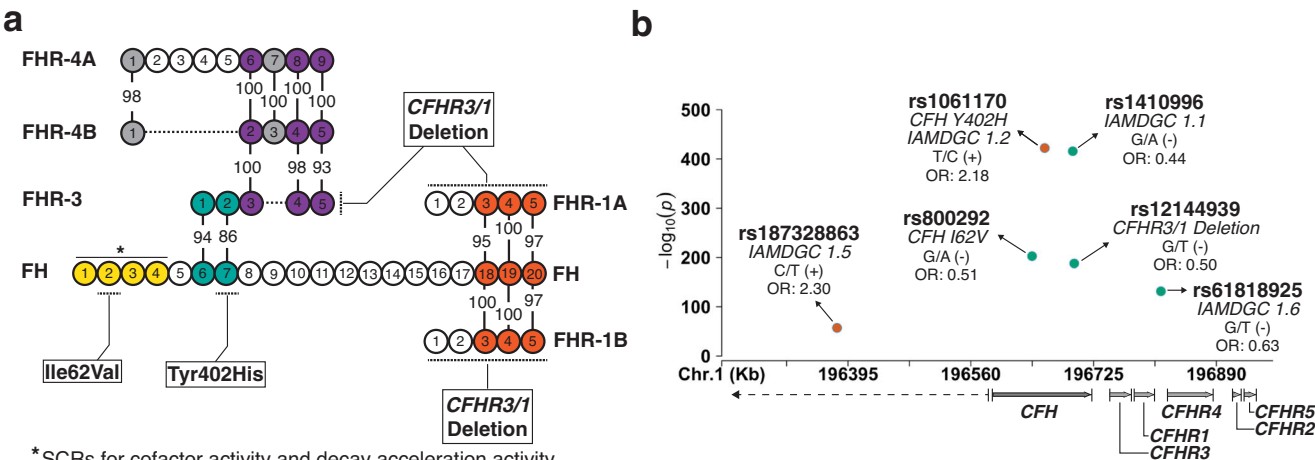

(1) Based in the IAMDGC cohort on the perfect proxy rs570618 ($r^2$ = 0.9914, D' = 1.0).
(2) Based in the IAMDGC cohort on rs6677604, another tag for the *CFHR3/1* deletion with minor allele A.

0.53], $p = 4.6 \times 10^{-202}$) or the *CFHR3/1* deletion (tagged by rs12144939, $OR_{IAMDGC}$: 0.50 [0.47; 0.52], $p = 7.1 \times 10^{-187}$). However, this variant does not tag perfectly these two forms of genetic protection. Approximately 1.9% of controls and 1% of cases in the Utah & Iowa cohort and 1.6% of controls and 0.8% of cases in the IAMDGC cohort have a haplotype associated with protection against AMD that carries the protective A allele at rs800292 in the absence of the minor allele at IAMDGC Locus 1.1 (see Supplementary Figs. 1 and 2). We therefore use the combination of rs800292 and rs12144939 in place of rs1410996. This choice

also ensures that the origin of the genetic protection among individuals protected against AMD can be readily identified. IAMDGC Locus 1.6 (rs6181825, $OR_{IAMDGC}$: 0.63 [0.60; 0.65], $p = 2.1 \times 10^{-131}$) is present on haplotypes associated with risk, protection or neutrality (defined as the absence of risk or protection[23]) for AMD (see Supplementary Fig. 3). In addition, the significance of its association with AMD is lost when conditioning regression models on rs1061170, rs800292 and rs12144939 (see Supplementary Figs. 2 and 4). Therefore, this variant does not differentiate disease susceptibility and was excluded from

**Fig. 1 | Single-variant, haplotype, and diplotype association analyses performed to refine AMD associations within the *CFH-CFHR5* extended region and to stratify human subjects and donor eyes by disease susceptibility, with associated functional effects. a** Schematic of the factor H (FH), factor H-like 1 (FHL-1), and factor H-related (FHR) family of proteins showing short consensus repeats (SCR) numbered from the N-terminus and percent amino acid similarities. The location of amino acid changes associated with FH/FHL-1 Y402H and FH/FHL-1 I62V is highlighted, as is the result of the *CFHR3/1* deletion. The schematic was minimally modified from Trends in Immunology, 36, Ózsi, M., Tortajada, A., Uzonyi, B., Goicoechea de Jorge, E. & Rodríguez de Córdoba, S., Factor H-related proteins determine complement-activating surfaces, 374–384, Copyright (2015), with permission from Elsevier[84]. In **b** displayed odds ratios (OR) were generated using the IAMDGC cohort (13,378 controls and 17,541 cases, see also Supplementary Table 2). Additional variant labels, major/minor alleles, and direction of effect on AMD susceptibility are also provided. **c** The combination of function-altering FH/FHL-1

Y402H and FH/FHL-1 I62V variants and the presence or absence of the *CFHR3/1* deletion yields four common haplotypes, with AMD associations ranging from protection to neutrality and risk. Odds ratios, confidence intervals (CI), and *p*-values (two-sided) were obtained using logistic regressions including AMD case/control status as the dependent variable and age, sex, and the first two genetic principal components for the IAMDGC cohort as covariates. Bonferroni correction for multiple testing of four haplotypes = 0.0125 (0.05/4). **d** Odds ratios and 95% CI for combinations of the four common haplotypes, generated independently with the IAMDGC (13,378 controls and 17,541 cases) and Utah & Iowa (1587 controls and 3200 cases) cohorts, reveal an AMD susceptibility continuum, with protective haplotypes mitigating risk haplotypes in a one-to-one manner. Odds ratios, 95% CI, and *p*-values (two-sided) were obtained using logistic regressions including AMD case/control status as the dependent variable and age, sex, and the first two genetic principal components for the IAMDGC cohort as covariates. Bonferroni correction for multiple testing of ten diplotypes = 0.005 (0.05/10).

further consideration. Finally, the risk allele for IAMDGC Locus 1.5 (rs187328863, $OR_{IAMDGC}$: 2.30 [2.08; 2.55], $p = 1.1 \times 10^{-57}$) resides exclusively on a less common haplotype (present in 5% of cases and 2.6% of controls in the Utah & Iowa cohort and in 3.5% of cases and 1.7% of controls in the IAMDGC cohort) containing the risk-associated C allele at rs1061170. Its contribution to AMD susceptibility is therefore accounted for by haplotypes including a risk allele at rs1061170. Altogether, these analyses indicate that three SNPs, consisting of the risk-associated rs1061170, the protection-associated rs800292 and rs12144939, a SNP tagging the *CFHR3/1* deletion haplotype, are the most likely causal AMD variants within the *CFH-CFHR5* extended region and are sufficient to describe the spectrum of common AMD susceptibility associated with the Chr1 AMD locus among individuals with European ancestry (see Fig. 1b and Supplementary Figs. 2 and 4).

The combination of rs1061170, rs800292, and rs12144939 yields four common haplotypes (frequency >1%) which capture 98.7% of control- and 99.3% of case-associated chromosomes in the Utah & Iowa cohort (98.8% and 99.4% in the IAMDGC cohort, respectively). Using a common neutral haplotype (present in ~20% of AMD cases and ~20% of controls in both Utah & Iowa and IAMDGC cohorts) as a reference[23] (see Fig. 1c), the most common haplotype based on the three variants carries the risk allele (C) at rs1061170 and no form of genetic protection for AMD; it is associated with an increased risk for developing disease ($OR_{IAMDGC}$: 1.58 [1.51; 1.65], $p = 2.0 \times 10^{-84}$). Two common protective haplotypes carry either the protective A allele at rs800292 (Prot-I62, $OR_{IAMDGC}$: 0.60 [0.57; 0.64], $p = 1.0 \times 10^{-70}$) or the *CFHR3/1* deletion (Prot-Del, $OR_{IAMDGC}$: 0.58 [0.45; 0.73], $p = 1.9 \times 10^{-76}$). Haplotype combinations (diplotypes) strongly influence AMD susceptibility (see Fig. 1d). Using the Neutral/Neutral diplotype (present in approximately 4% of cases and 4% of controls in both Utah & Iowa and IAMDGC cohorts) as a reference, the Risk/Risk diplotype confers an increased risk for developing AMD ($OR_{IAMDGC}$: 2.87 [2.5; 3.3], $p = 1.5 \times 10^{-57}$). In contrast, combinations including risk and protective haplotypes are generally protected against AMD, with protective haplotypes mitigating the risk haplotype in a one-to-one manner.

### Genetic determinants of FHR-4 levels in plasma
To dissect the association between variation in FHR-4 and AMD susceptibility, we sought to identify independent genetic modulators of FHR-4 levels. The *CFHR4* gene is not expressed in ocular tissues[52] and plasma FHR-4 is predominantly produced in the liver[53]. Variants independently associated with *CFHR4* transcript levels (*cis*-expression quantitative loci, *cis*-eQTL) were identified using genotype and liver gene expression of 183 donors (129 males, 54 females) with European ancestry (median age 56 IQR 14.5) from the Genotype-Tissue Expression Project (GTEx)[54], version 8. The two common independent *CFHR4 cis*-eQTLs identified through conditional analysis, rs559637118 ($\beta = -0.55$, $p = 1.0 \times 10^{-08}$) and rs1830959 ($\beta = -0.48$, $p = 1.7 \times 10^{-05}$), are associated with reduced *CFHR4* transcript levels. The *CFH* intron 10

variant rs559637118 is a perfect proxy for rs1410996 ($r^2 = 1.0$; $D' = 1.0$) among GTEx liver donors. The *CFHR2* intron 1 variant rs1830959 is in perfect LD with the intergenic rs7531555 ($r^2 = 1.0$; $D' = 1.0$) among these donors (see Supplementary Table 3). Independent FHR-4 protein quantitative trait loci (pQTL, either in *cis* or *trans*) were identified by mining large plasma genome-proteome-wide association studies performed among individuals with European ancestry[55,56]. This search yielded three independent variants, rs10494745 ($\beta = -0.985$, $p = 8.0 \times 10^{-529}$ in the analysis by Pietzner et al.[55] and $\beta = -0.899$, $p = 0$ in the study by Gudjonsson et al.[56]); rs61818956 ($\beta = -0.55$, $p = 1.0 \times 10^{-08}$ in Pietzner et al.[55]) and two variants (rs4915363 and rs7413610) in strong LD with rs7531555 ($r^2 = 0.953$, $D' = 0.984$ and $r^2 = 0.912$, $D' = 0.958$, respectively), with $\beta = 0.785$ and $p = 1.3 \times 10^{-662}$ in Pietzner et al.[55] and $\beta = -0.678$, $p = 1.7 \times 10^{-246}$ in Gudjonsson et al.[56] (see Fig. 2a and Supplementary Table 4). The missense *CFHR4* variant rs10494745 (p.Gly553Glu), which is associated with reduced FHR-4 levels in plasma, was not identified as a significant eQTL for *CFHR4* among GTEx liver donors. The minor allele of the *CFHR4* intron 9 variant rs61818956 is associated with elevated FHR-4 levels in plasma. No significant *CFHR4 trans*-pQTLs were identified in plasma-proteome-wide association studies.

### Development and validation of a fit-for-purpose FHR-4 ELISA
To confirm the effect of liver QTLs in plasma and ocular samples, we developed an in-house, fit-for-purpose ELISA capable of detecting total FHR-4 protein. A fit-for-purpose study was performed to establish the robustness of the FHR-4 ELISA and to determine cross-reactivity to all complement factor H-related family proteins. Further validation of ELISA specificity and sensitivity in human plasma and serum samples was performed using subjects stratified based on the presence of genetic deletions of *CFHR1/4*. These subjects were selected from the Utah & Iowa cohort (57 heterozygous samples), and from samples collected from a subset of individuals from a Rapa Nui cohort (15 heterozygous samples and 8 homozygous deletion samples) with a high frequency of the *CFHR1/4* deletion (approximately 20%) held at the Sharon Eccles Steele Center for Translational Medicine, University of Utah. Among individuals from the Utah & Iowa cohort, a significant reduction in plasma FHR-4 was observed in heterozygous subjects ($p < 0.001$). In Rapa Nui serum samples, FHR-4 levels were significantly lower among heterozygotes or homozygotes for the *CFHR1/4* deletion as compared to individuals carrying no deletion ($p < 0.001$); see Supplementary Fig. 5.

### Confirmation of QTL effects in plasma samples
FHR-4 protein concentration was measured in plasma collected from 486 individuals from the Utah & Iowa cohort (median age 75.0 IQR 13.0) with no AMD (235 subjects), early/intermediate AMD (120 subjects) or late AMD (131 subjects); see Supplementary Table 5 for detailed demographics. No associations were observed between

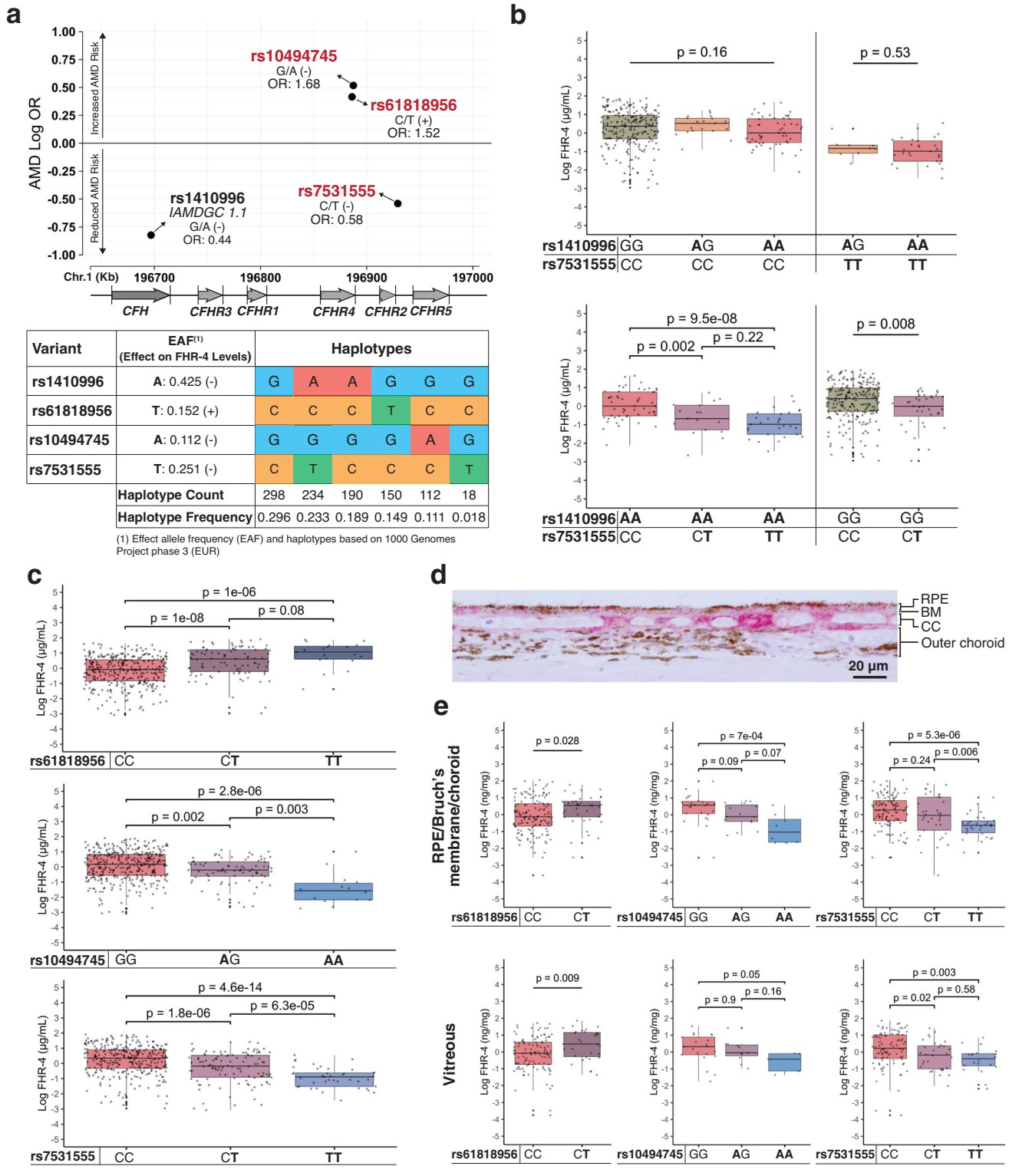

FHR-4 levels and age (*p* = 0.3), AMD status (*p* = 0.57) or AMD stage of severity (*p* = 0.62); see Supplementary Fig. 6. We observed that the minor alleles at rs10494745 (A), rs7531555 (T) and rs1410996 (A) were associated with a significant reduction of FHR-4 levels in plasma (*p* < 0.001); see Fig. 2b, c and Supplementary Fig. 7. The minor allele at rs61818956 (T) was associated with significantly higher levels of FHR-4 in plasma (*p* < 0.001). Both elevated and reduced levels of plasma FHR-4 were independent of AMD status, as shown in Supplementary Fig. 8.

### The effect of rs1410996 (IAMDGC 1.1) on FHR-4 plasma levels is attributable to rs7531555

Haplotype analyses performed among individuals with European ancestry from the 1000 Genomes project phase 3 (1000 G EUR)[57] (Fig. 2a) and independently among controls of the Utah & Iowa and IAMDGC cohorts (Supplementary Fig. 9) confirmed that the minor allele at rs10494745 (A) or rs61818956 (T) did not overlap with the FHR-4 lowering alleles at rs7531555 (T) or rs1410996 (A), nor did they ever share a chromosome with each other. However, approximately 23% of

**Fig. 2 | Genetic modulators of FHR-4 levels in plasma, RPE/Bruch's membrane/choroid, and vitreous humor. a** Graph showing the AMD odds ratios (OR), generated using the IAMDGC cohort (13,378 controls and 17,541 cases, see also Supplementary Tables 3 and 4), of quantitative trait loci (QTLs) identified through expression QTL (rs1410996, rs7531555) or protein QTL (rs61818956, rs10494745, rs7531555) analyses as a function of location on chromosome 1. The major/minor alleles and effect of the minor alleles on FHR-4 variation are also shown. Independent QTLs, highlighted in red, were identified through haplotype analyses performed using 1000 Genomes Project phase 3 (1000 G EUR) subjects (shown in the panel) and controls from the Utah & Iowa and IAMDGC cohorts (Supplementary Fig. 9). **b** Box plots of log-transformed and centered plasma FHR-4 levels by rs1410996 (IAMDGC 1.1) and rs7531555 genotypes (404 independent biological samples). Analyses of genetic combinations indicate that the expression QTL effect associated with rs1410996 is entirely attributable to rs7531555. **c** Box plot showing the distribution of log-transformed and centered plasma FHR-4 levels by rs61818956, rs10494745, and rs7531555 genotypes (486 independent biological samples). **d** FHR-4 immunohistochemistry in retinal pigment epithelium (RPE)/

Bruch's membrane/choroid tissues from an eye donor obtained using anti-FHR-4 antibody and warp red chromagen shows intense staining in Bruch's membrane (BM) and between pillars of the choriocapillaris (CC). An image of a negative control can be found in Supplementary Fig. 12. **e** Variations in log-transformed and centered FHR-4 levels measured in RPE/Bruch's membrane/choroid and vitreous lysates with the three independent rs61818956, rs10494745, and rs7531555 genotypes (229 independent biological samples, see Supplementary Table 6). The directional effect of QTLs in ocular samples is consistent with that observed in plasma samples. In **b**, **c**, and **e**, in all box plots the horizontal center lines correspond to the medians of the log-transformed and centered FHR-4 distribution and the boxes delineate the 25th/75th percentile. The vertical solid lines represent the full range of the log-transformed and centered FHR-4 distribution in each group. Dots beyond this line indicate potential outliers. Associations with log-transformed and centered FHR-4 levels were assessed using the Kruskal−Wallis test. Post-hoc pairwise comparisons were performed using the Conover−Iman test. All $p$-values are two-sided and adjusted for multiple testing using Bonferroni corrections. Source data for these plots are provided as a Source Data file.

chromosomes from 1000 G EUR participants (22% of controls from the Utah & Iowa and IAMDGC cohorts; see Supplementary Fig. 9) carry the allele associated with lower FHR-4 levels at both rs7531555 (T) and rs1410996 (A), which indicates that the effect associated with these two variants may not be independent. To further dissect the respective effect of these QTLs, we compared FHR-4 levels among subjects stratified by rs7531555 and rs1410996 genotypes. We found that FHR-4 levels did not vary significantly with the level-reducing allele (A) at rs1410996 when rs7531555 genotype was kept constant (see Fig. 2b). Conversely, FHR-4 levels were significantly associated with rs7531555 genotype even when rs1410996 was kept constant. Specifically, among subjects homozygous for the alternative allele at rs1410996 (G), the FHR-4 reducing allele at rs7531555 (T) was associated with significantly lower FHR-4 levels ($p = 0.008$). The same effect was observed among samples with an AA genotype at rs1410996 (rs7531555 CT vs CC: $p = 0.002$; rs7531555 TT vs CC: $p = 9.5 \times 10^{-08}$). These analyses therefore indicate that rs1410996 is not an independent QTL for *CFHR4* mRNA expression, as its effect on FHR-4 is exclusively attributable to the rs7531555 T allele.

## Genetic modulators of FHR-4 influence levels in RPE/Bruch's membrane/choroid and vitreous samples

Altogether, rs61818956, rs10494745, and rs7531555 describe three independent genetic variations that influence FHR-4 plasma levels. We sought to determine if these variants also influenced the levels of this protein in ocular compartments, including the primary location of AMD pathology, the RPE/Bruch's membrane/choroid interface, and in the vitreous humor. Immunohistochemistry of human donor eyes indicated that FHR-4 was present in Bruch's membrane and between pillars of the choriocapillaris (see Fig. 2d). We measured FHR-4 levels in both RPE/Bruch's membrane/choroid tissue and vitreous humor lysates collected from 229 donor eyes (median age 77.0 IQR 13.0) using the FHR-4 ELISA; see Supplementary Table 6 for detailed demographics. FHR-4 levels in the RPE/Bruch's membrane/choroid tissue and vitreous humor of the same donor (136 eyes) were significantly correlated (Pearson's correlation coefficient: 0.58, $p < 0.001$). Overall, the effect of rs61818956, rs10494745, and rs7531555 SNPs in ocular compartments was similar to that observed in plasma. Specifically, FHR-4 levels were significantly elevated among donors carrying a T allele at rs61818956 in both RPE/Bruch's membrane/choroid (CC vs CT, $p = 0.028$) and vitreous humor (CC vs CT, $p = 0.009$); see Fig. 2d. The FHR-4 reducing allele at rs10494745 and rs7531555 were both associated with lower FHR-4 levels in RPE/Bruch's membrane/choroid tissue (rs10494745 AA vs GG: $p = 7.0 \times 10^{-04}$; rs7531555 TT vs CC: $p = 5.3 \times 10^{-06}$) and vitreous humor (rs10494745 AA vs GG: $p = 0.05$; rs7531555 TT vs CC: $p = 0.003$).

## Associations of CFHR4 QTLs with AMD are attributable to rs1061170, rs800292, and the CFHR3/1 deletion haplotype

Independent *CFHR4* QTLs influence the concentration of FHR-4 in plasma and at the primary location of AMD-associated pathology. Single-variant association analyses performed using the Utah & Iowa cohort and validated against the IAMDGC cohort indicated that the minor alleles at rs10494745 (A) and rs61818956 (T), which have opposing effects on FHR-4 levels, were both associated with an increased risk for AMD ($OR_{IAMDGC}$: 1.68 [1.60; 1.77], $p = 7.8 \times 10^{-88}$ and $OR_{IAMDGC}$: 1.52 [1.45; 1.58], $p = 7.0 \times 10^{-82}$, respectively), while the FHR-4 lowering allele at rs7531555 (T) is associated with protection against AMD ($OR_{IAMDGC}$: 0.58 [0.56; 0.61], $p = 2.7 \times 10^{-135}$); see Fig. 2a and Supplementary Tables 3 and 4. To explain these associations, we performed independent haplotype analyses on AMD status among the Utah & Iowa and IAMDGC cohorts, see Fig. 3a. We found that associations between rs10494745 and rs61818956 and increased AMD risk were entirely attributable to the risk-associated allele at rs1061170 (C) that encodes FH/FHL-1 H402. The FHR-4 reducing allele at rs10494745 (A) resides almost exclusively on a haplotype carrying the risk-associated C allele at rs1061170 ($D' = 0.96$ and $r^2 = 0.21$). The FHR-4 increasing allele at rs61818956 (T) exists primarily as part of a risk-associated haplotype (H5 on Fig. 3a, 10.1% of 1000 G EUR chromosomes and 11.2% and 12.3% of chromosomes from Utah & Iowa and IAMDGC controls, respectively) that also carries the C allele at rs1061170. When it is present without the C at rs1061170, it is on a neutral haplotype (H7, 4.3% of 1000 G EUR chromosomes and 3.4% and 4.4% of chromosomes from Utah & Iowa and IAMDGC controls, respectively). The protective effect associated with the FHR-4 lowering allele at rs7531555 (T) is explained by the fact that approximately 20% of chromosomes (20.1% from the 1000 G EUR and 19.7% of chromosomes from Utah & Iowa and IAMDGC controls, respectively) contain both the T allele at rs7531555 and the protective A allele at rs800292. Furthermore, no protective effect is offered by the T allele at rs7531555 when it is present without protection coming from the already established A allele at rs800292 or T allele at rs12144939 loci. In fact, the addition of any of the three FHR-4 QTL effect alleles has no influence on the overall AMD susceptibility status (risk, neutral or protection) of the 3-SNP haplotype containing rs800292, rs1061170 and rs12144939 (see Figs. 1c and 3a).

## AMD-associated variants do not independently modulate FHR-4 plasma levels

To further assess if FHR-4 levels could modify AMD susceptibility, we determined if AMD-associated variants/haplotypes had an independent effect on protein levels. FHR-4 levels are significantly lower among individuals homozygous for the Prot-I62 haplotype as

**a**

| Haplotype (Effect) | rs800292 | rs1061170(1) | rs12144939(2) | rs61818956 | rs10494745 | rs7531555 | 1000 G EUR Frequency | Utah & Iowa Cohort 1,587 controls; 3,200 cases |  |  |  |  | IAMDGC Cohort 13,378 controls; 17,541 cases |  |  |  |  | Effect on FHR-4 Levels |
|---|---|---|---|---|---|---|---|---|---|---|---|---|---|---|---|---|---|---|
|  |  |  |  |  |  |  |  | Frequency Controls | Frequency Cases | Score Statistic | p-value* | OR (95 %CI) | p-value | Frequency Controls | Frequency Cases | Score Statistic | p-value* | OR (95 %CI) | p-value |  |
| H1 Prot-I62 | A | T | G | C | G | T | 0.214 | 0.201 | 0.118 | -10.68 | 1.3e-26 | 0.61 [0.52; 0.73] | 9.1e-9 | 0.197 | 0.116 | -27.57 | 2.7e-167 | 0.59 [0.56; 0.63] | 7.7e-59 | Reduced |
| H2 Prot-Del | G | T | T | C | G | C | 0.167 | 0.181 | 0.097 | -11.94 | 7.1e-33 | 0.53 [0.45; 0.64] | 1.7e-11 | 0.183 | 0.103 | -27.83 | 1.8e-170 | 0.57 [0.53; 0.61] | 6.0e-66 | Baseline |
| H3 Neutral | G | T | G | C | G | C | 0.145 | 0.156 | 0.153 | -1.23 | 0.22 | 1.0 (reference) | - | 0.140 | 0.142 | -1.55 | 0.12 | 1.0 (reference) | - | Baseline |
| H4 Risk | G | C | G | C | G | C | 0.135 | 0.147 | 0.210 | 7.96 | 1.7e-15 | 1.53 [1.28; 1.82] | 4.2e-6 | 0.131 | 0.186 | 19.35 | 1.9e-83 | 1.44 [1.35; 1.54] | 3.1e-30 | Baseline |
| H5 Risk | G | C | G | T | G | C | 0.101 | 0.112 | 0.177 | 8.86 | 8.1e-19 | 1.72 [1.44; 2.05] | 1.3e-9 | 0.123 | 0.191 | 23.39 | 5.7e-121 | 1.59 [1.49; 1.70] | 8.6e-48 | Elevated |
| H6 Risk | G | C | G | C | A | C | 0.109 | 0.093 | 0.151 | 7.77 | 7.8e-15 | 1.69 [1.40; 2.04] | 3.1e-8 | 0.100 | 0.159 | 20.74 | 1.4e-95 | 1.60 [1.50; 1.72] | 3.7e-44 | Reduced |
| H7 Neutral | G | T | G | T | G | C | 0.043 | 0.034 | 0.033 | -0.06 | 0.95 | 1.05 [0.77; 1.44] | 0.75 | 0.044 | 0.040 | -1.91 | 0.06 | 0.95 [0.86; 1.05] | 0.29 | Elevated |
| H8 Prot-I62 | A | T | G | C | G | C | 0.024 | 0.024 | 0.015 | -3.03 | 2.4e-3 | 0.70 [0.48; 1.02] | 3.5e-2 | 0.026 | 0.014 | -11.00 | 3.5e-28 | 0.56 [0.49; 0.65] | 2.1e-17 | Baseline |
| H9 Neutral | G | T | G | C | G | T | 0.020 | 0.014 | 0.013 | -0.47 | 0.64 | 1.0 [0.62; 1.60] | 0.97 | 0.018 | 0.017 | -0.76 | 0.45 | 0.94 [0.80; 1.10] | 0.36 | Reduced |
| H10 Prot-Del | G | T | T | C | G | T | 0.009 | 0.010 | 0.006 | -3.12 | 1.8e-3 | 0.61 [0.41; 0.93] | 5.0e-2 | 0.013 | 0.008 | -7.67 | 1.7e-14 | 0.55 [0.45; 0.67] | 3.9e-11 | Reduced |
| H11 Risk | G | C | G | C | G | T | 0.008 | 0.010 | 0.014 | 2.19 | 0.03 | 1.66 [0.99; 2.78] | 4.5e-2 | 0.008 | 0.015 | 8.16 | 3.3e-16 | 1.95 [1.60; 2.38] | 2.7e-14 | Reduced |

(1) Based in the IAMDGC cohort on the perfect proxy rs570618 ($r^2$ = 0.9914, D' = 1.0).
(2) Based in the IAMDGC cohort on rs6677604, another tag for the *CFHR3/1* deletion with minor allele A.
*Based on the $\chi^2$ test with one degree of freedom.

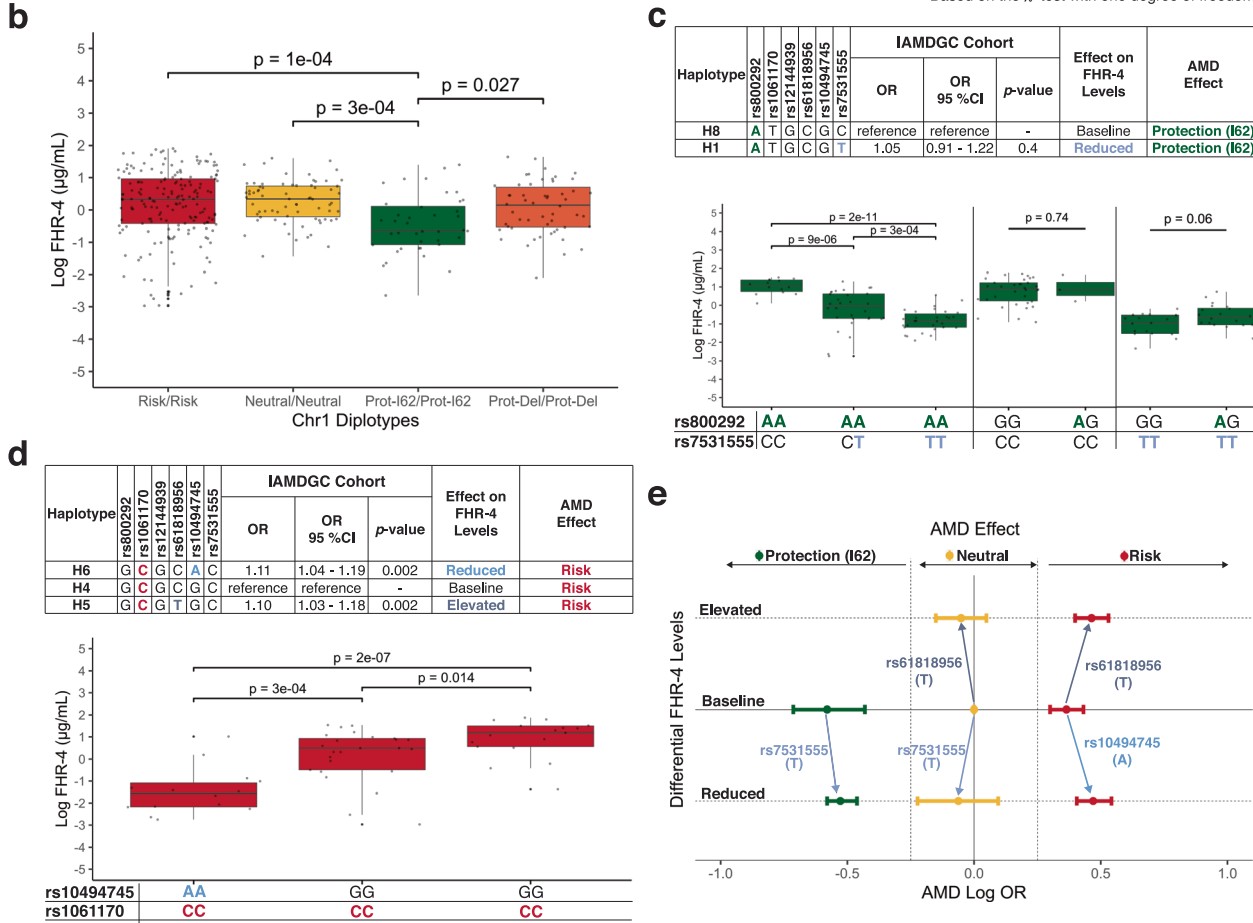

compared to subjects with Risk/Risk ($p = 1.0 \times 10^{-04}$), Neutral/Neutral ($p = 3.0 \times 10^{-04}$) and Prot-Del/Prot-Del ($p = 0.027$) diplotypes (see Fig. 3b). Examination of genotype combinations demonstrates that these differences are driven exclusively by rs7531555. FHR-4 levels do not vary significantly with the protective A allele at rs800292 and a C allele at rs7531555 (referred to as baseline) but are reduced with a T allele at rs7531555 (see Fig. 3c). In contrast, among subjects homozygous for the Prot-I62 genotype (AA at rs800292), the T allele at rs7531555 is associated with significantly lower FHR-4 levels (CT vs CC: $p\,9.0 \times 10^{-06}$; TT vs CC: $p\,23.0 \times 10^{-11}$). We also confirmed that the effects of rs61818956 and rs10494745 SNPs on FHR-4 levels are independent from rs1061170 in subjects with Risk/Risk diplotypes (see

**Fig. 3 | Dissection of associations between FHR-4 levels and AMD susceptibility.**
**a** Haplotype analyses based on the AMD-associated risk allele C at rs1061170, protective allele A at rs800292, and protective *CFHR3/1* deletion tagging allele T at rs12144939 and independent quantitative trait loci (QTL) for *CFHR4* rs61818956, rs10494745, and rs7531555. Odds ratios (OR), confidence intervals (CI), and *p*-values (two-sided) were obtained using logistic regressions including AMD case/control status as the dependent variable and age, sex, and the first two genetic principal components for the IAMDGC cohort as covariates. Bonferroni correction for multiple testing of 11 haplotypes = 0.0045 (0.05/11). The differential effects on FHR-4 levels associated with each haplotype (elevated, baseline, or reduced) is indicated.
**b** Box plot showing variations in log-transformed and centered FHR-4 levels with the four common AMD homozygous Chr1 diplotypes (Risk, Neutral, Prot-I62, and Prot-Del) (346 independent biological samples). **c** Comparison of Prot-I62 haplotypes with and without the effect allele (T) at rs7531555 generated using the IAMDGC cohort (13,378 controls and 17,541 cases) along with a box plot showing variations in log-transformed and centered FHR-4 levels among subjects stratified by rs800292 and rs7531555 diplotypes (149 independent biological samples).
**d** Comparison of Risk haplotypes with and without the effect allele at rs10494745 and rs61818956 generated using the IAMDGC cohort (13,378 controls and 17,541

cases) along with a box plot showing variations in log-transformed and centered FHR-4 levels among subjects with Risk/Risk diplotypes stratified by rs10494745 (AA or GG) and rs61818956 (CC or TT) diplotypes (56 independent biological samples). **e** Schematic describing the effect of reduced or elevated FHR-4 levels on AMD susceptibility by haplotype group. Log-transformed OR and 95% CI were generated using the IAMDGC cohort (13,378 controls and 17,541 cases). In **b**, **c**, and **d**, OR, 95% CI and *p*-values (two-sided) for comparisons between haplotypes were obtained using logistic regressions including AMD case/control status as the dependent variable and age, sex, and the first two genetic principal components as covariates. In all box plots the horizontal center lines correspond to the medians of the log-transformed and centered FHR-4 distribution and the boxes delineate the 25th/75th percentile. The vertical solid lines represent the full range of the log-transformed and centered FHR-4 distribution in each group. Dots beyond this line indicate potential outliers. Associations with log-transformed and centered FHR-4 were assessed using the Kruskal–Wallis test. Post-hoc pairwise comparisons were performed using the Conover–Iman test. All *p*-values are two-sided and adjusted for multiple testing using Bonferroni corrections. Source data for these plots are provided as a Source Data file.

Fig. 3d). Among these individuals, the A allele at rs10494745 is associated with significantly lower FHR-4 levels (AA vs GG: $p = 3.0 \times 10^{-04}$) while the T allele at rs61818956 results in significantly higher FHR-4 levels (TT vs CC: $p = 0.014$). Therefore, variants and haplotypes describing all common susceptibility for AMD at the Chr1 AMD locus have no independent effect on FHR-4 levels. And, conversely, the three independent QTLs rs61818956, rs10494745, and rs7531555 modulate FHR-4 levels in plasma and at the primary location of AMD-associated pathology independently from variants and haplotypes associated with AMD.

### FHR-4 plasma and ocular levels do not influence AMD susceptibility or disease progression
We examined the differential effect of FHR-4 variation on AMD susceptibility in the two large, independent Utah & Iowa and IAMDGC case/control cohorts. To do so, we assessed associations of haplotypes based on the AMD-associated variants rs1061170, rs800292 and rs12144939 and *CFHR4* QTLs (associated with reduced, baseline or elevated FHR-4 levels) with AMD status (see Fig. 3a). AMD odds ratios in the Prot-I62 and Neutral haplotype groups did not differ significantly between subjects with reduced FHR-4 levels as compared to those with baseline concentrations (see Fig. 3c and Supplementary Fig. 10). Among chromosomes with a neutral haplotype, no significant effect on AMD odds ratios was detected for elevated FHR-4 levels (Supplementary Fig. 10). Among chromosomes with a risk haplotype, both reduced (A at rs10494745) and elevated (T at rs61818956) FHR-4 levels are associated with a 1.1-fold increase in AMD odds ratios as compared to baseline ($p = 0.002$ for both variants; see Fig. 3d). Altogether, genetic variation in FHR-4 levels do not influence the protection against AMD conferred by the Prot-I62 haplotype, the risk for AMD associated with the Risk haplotype or the lack of protection or risk associated with the Neutral haplotype, see Fig. 3e.

Since both elevated and reduced plasma and ocular tissue levels of FHR-4 coincide with a higher risk for AMD, variation in FHR-4 is unlikely to mitigate AMD risk. However, factors influencing AMD susceptibility may differ from those affecting its progression course. Therefore, we assessed if variations in FHR-4 levels impacted the course of AMD among patients with disease driven primarily by AP dysregulation. To do so, we analyzed longitudinal clinical data collected from 317 patients with a Risk/Risk diplotype and no risk alleles at the Chr10 AMD locus[16]. To ensure that comparisons were performed between groups of individuals with the largest difference in FHR-4 levels, we selected patients on the basis on homozygosity at rs61818956 (TT, FHR-4↑ group) and rs10494745 (AA, FHR-4↓ group), with the reference group designated FHR-4↔ (CC at rs61818956, GG at

rs10494745). We only included patients with at least one eye at risk of conversion (presenting clinical signs of early or intermediate AMD) to late-stage AMD (cRORA or neovascular AMD) at the first visit. A total of 81 patients (131 eyes) met these inclusion criteria (see Fig. 4a). Over the course of follow-up, the frequency of conversions to late-stage AMD did not differ significantly between the FHR-4↑, FHR-4↔ and FHR-4↓ groups (see Fig. 4b). We observed no significant differences in the median age of first recorded conversion to late-stage AMD (see Fig. 4c). A Cox proportional hazards (CoxPH) model for earliest conversion (first eye to convert) to late AMD adjusted for age and AMD severity at first visit did not reveal any significant differences in conversion time between patients with reduced or elevated levels of FHR-4 as compared to those with baseline levels. Mixed-effect CoxPH models adjusted for the same covariates and including a frailty factor to account for correlations between eyes of the same patient did not yield any significant associations (see Fig. 4d). This analysis indicates that in addition to AMD susceptibility, variation in FHR-4 levels does not affect the course of progression of this disease.

### Discussion
Using liver gene expression data, the two largest plasma genome-proteome-wide association studies to date[55,56] and a large number of human plasma and ocular tissue samples, we identified three independent genetic modulators of FHR-4 levels in plasma. We demonstrated their effect on FHR-4 levels in ocular compartments, including the primary location of AMD-associated pathology, the RPE/Bruch's membrane/choroid interface. Analyses performed using two distinct cohorts composed of a combined 14,965 controls and 20,741 cases revealed that these three genetic variants were not independently associated with AMD. In fact, all associations between AMD-associated variants and FHR-4 levels are entirely explained by these QTLs. These refinements allowed us to show that genetically driven variation in FHR-4 levels do not influence AMD susceptibility or its course of progression.

Genetic modulation of FHR-4 levels associated with rs61818956, rs10494745, and rs7531555 was detected in plasma samples from a large number of human subjects, with effects that were independent of AMD status. Out of these three independent QTLs, only rs7531555 was ascertained as both an independent *cis*-eQTL and *cis*-pQTL for *CFHR4*. The missense *CFHR4* variant rs10494745 and the *CFHR4* intron 9 variant rs61818956 were only identified as *cis*-pQTLs. No pQTLs in *trans* were found. A previous study identified rs10494745 and a proxy for rs1410996 (rs10922108, $r^2 = 1.0$ and $D' = 1.0$) as pQTLs for *CFHR4*[42]. Another investigation only ascertained proxies for rs7531555[44] (rs4085749, $r^2 = 1.0$ and $D' = 1.0$; and rs12047098, $r^2 = 0.826$ and

**a**

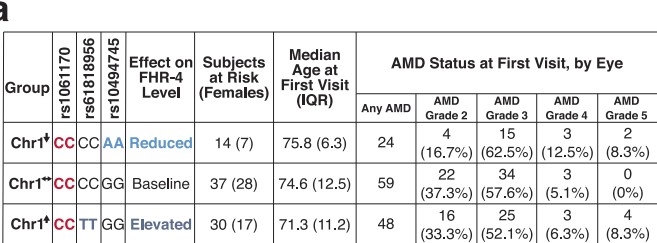

| Group | rs1061170 | rs61818956 | rs10494745 | Effect on FHR-4 Level | Subjects at Risk (Females) | Median Age at First Visit (IQR) | AMD Status at First Visit, by Eye | | | | | |
|---|---|---|---|---|---|---|---|---|---|---|---|---|
| | | | | | | | Any AMD | AMD Grade 2 | AMD Grade 3 | AMD Grade 4 | AMD Grade 5 | |
| Chr1↓ | CC | CC | AA | Reduced | 14 (7) | 75.8 (6.3) | 24 | 4 (16.7%) | 15 (62.5%) | 3 (12.5%) | 2 (8.3%) | |
| Chr1•• | CC | CC | GG | Baseline | 37 (28) | 74.6 (12.5) | 59 | 22 (37.3%) | 34 (57.6%) | 3 (5.1%) | 0 (0%) | |
| Chr1↑ | CC | TT | GG | Elevated | 30 (17) | 71.3 (11.2) | 48 | 16 (33.3%) | 25 (52.1%) | 3 (6.3%) | 4 (8.3%) | |

AMD Grade 2 - Early AMD
AMD Grade 3 - Intermediate AMD, without pigment epithelial detachment larger than 1000µm
AMD Grade 4 - Intermediate AMD, with pigment epithelial detachment larger than 1000µm
AMD Grade 5 - Presence of incomplete retinal pigment epithelial and outer retinal atrophy (iRORA)

**b**

| Eyes | Chr1↓ | Chr1•• | Chr1↑ | p-value |
|---|---|---|---|---|
| At risk at first visit | 24 | 59 | 48 | |
| No recorded conversions | 15 (62.5%) | 45 (76.3%) | 33 (68.8%) | p = 0.4 |
| Recorded conversions | 9 (37.5%) | 14 (23.7%) | 15 (31.3%) | (X² = 1.8)(3) |
| Type of conversion | | | | |
| cRORA(1) | 5 (55.6%) | 6 (42.9%) | 11 (73.3%) | |
| Neovascular AMD | 2 (22.2%) | 7 (50%) | 4 (26.7%) | |
| Unknown(2) | 2 (22.2%) | 1 (7.1%) | 0 | |

(1) Complete retinal pigment epithelial and outer retinal atrophy
(2) Nature of earliest conversion (cRORA/neovascular AMD) could not be reliably determined
(3) Based on the χ² test with 2 degrees of freedom

**c**

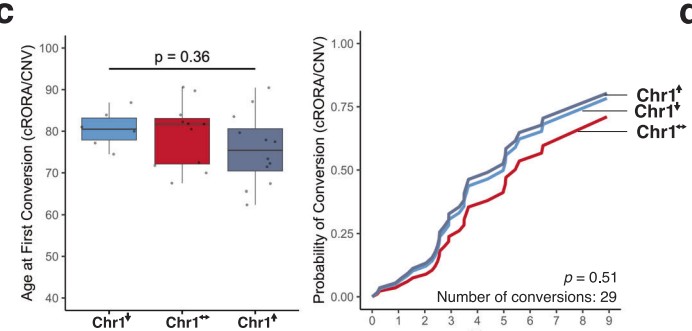

**d**

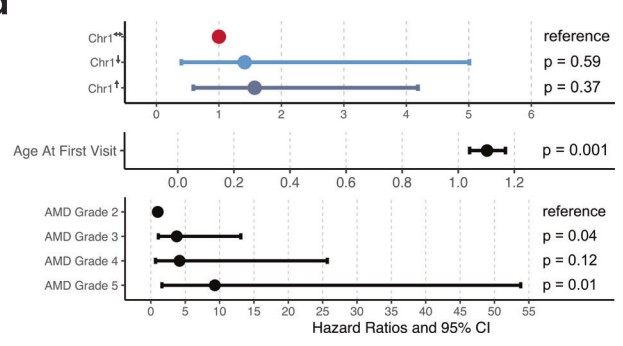

**Fig. 4 | Effect of variations in FHR-4 plasma and ocular levels on disease progression in AMD driven by rs1061170 risk. a** Patients with a Risk/Risk diplotype and no risk alleles at the Chr10 AMD locus were selected based on genotype at rs61818956 (TT, FHR-4↑ group) and rs10494745 (AA, FHR-4↓ group), with the reference group labeled FHR-4•• (CC at rs61818956, GG at rs10494745). **b** Table showing the number and frequency of conversions recorded in each eye that met the inclusion criteria. Differences between groups were assessed using a chi-squared test. *P*-values are two-sided. **c** Box plots showing the median age of first recorded conversion to late AMD (cRORA or neovascular AMD) and conversion curves for earliest conversion generated using Cox proportional hazard regression models adjusted for age and AMD severity at first visit (81 patients, see **a**). In the box plot, the horizontal center lines correspond to the medians and the boxes delineate the 25th/75th percentile. The vertical solid lines represent the full range of the age distribution in each group. Dots beyond this line indicate potential outliers. Association with age at first conversion was assessed using the Kruskal–Wallis test. Associations with time to conversion were determined using the log-rank test. All *p*-values are two-sided. **d** Hazard ratios and 95% confidence interval (CI) generated using mixed-effect Cox proportional hazard regression models adjusted for age and AMD severity at first visit and including a frailty term to account for correlations between eyes from the same patient (131 eyes from 81 patients, see **a**). All *p*-values are two-sided.

$D' = 0.949$) as lead *CFHR4* pQTLs (see Supplementary Table 7). In both studies, the directions of effect were consistent with our findings. We demonstrated the effect of rs61818956, rs10494745, and rs7531555 on FHR-4 levels in the primary location of AMD pathology, the RPE/Bruch's membrane/choroid interface, as well as in the vitreous humor of human donor eyes. Since *CFHR4* mRNA is not expressed in ocular tissue, the correlation between plasma and tissue levels is not surprising and is analogous to liver-produced systemic FHR-1 and FHR-3 proteins gaining access to ocular compartments.

Previous studies sought to establish that FHR-4 levels influence AMD susceptibility by demonstrating an overlap between SNPs driving genetic variations in plasma FHR-4 and variants and haplotypes associated with AMD at the Chr1 AMD locus (see Supplementary Table 8 for a detailed comparison with these studies and Supplementary Note 1 for an extended discussion on similarities and discrepancies with these investigations). Our analyses show that the genetic variants influencing FHR-4 levels are separate from the ones determining AMD susceptibility at the Chr1 locus. Associations between AMD-associated variants/haplotypes and FHR-4 levels are due to linkage disequilibrium rather than causation. Associations with FHR-4 levels reported with the AMD-associated rs1410996 (IAMDGC 1.1) and rs1061170 (*CFH* Y402H, IAMDGC 1.2)[42–44,58] are attributable to rs7531555 and to the combination of rs61818956 and rs10494745, respectively. Because they have an opposite differential effect on FHR-4 levels and are present on AMD risk haplotypes, the frequencies of the minor alleles at rs61818956 and rs10494745 are likely to strongly influence the variability in FHR-4 levels among groups of individuals carrying high-risk haplotypes at the Chr1 AMD locus and among patients with AMD[42–44]. Significantly lower

FHR-4 that we and others[42–44] observed among individuals carrying the protective A allele at rs800292, which is more frequent among controls, as compared to subjects carrying the risk-associated C allele at rs1061170, are likely to be driven by the level-reducing T allele at rs7531555. Altogether, the presence of opposing QTLs on AMD risk haplotypes, which are more common among AMD patients, combined with the presence of the level-reducing T at rs7531555 on protective haplotypes, which are more common among controls, may explain previous findings, established using small sample sizes, that FHR-4 levels were elevated among individuals with AMD[43]. Since the FHR-4 QTLs that we identified were established from multiple datasets and genome-wide analysis methodologies, our findings are unlikely to be due to incomplete variant discovery. Further, it is likely that all causal SNPs within the Chr1 AMD locus have already been ascertained or are in tight LD with those that have already been discovered. The absence of overlap between variants affecting FHR-4 levels and AMD status that we report is therefore robust and demonstrates that *CFHR4* is unlikely to play a role in AMD disease mechanisms.

The stratification of subjects and donor eyes applied in this study was based upon a set of variants, haplotypes and diplotypes describing the entire spectrum of common susceptibility to AMD, which ranges from protection to neutrality and risk. We used the combination of rs800292 and the *CFHR3/1* deletion tagging SNP rs12144939 in place of rs1410996 (IAMDGC 1.1) since this variant does not tag perfectly these two common forms of genetic protection. We previously showed[23] using the Utah & Iowa cohort only that rs1410996 may have been identified as a most likely causal variant at the Chr1 AMD locus by the IAMDGC[19] because it tags two independent protective haplotypes

carrying an A allele at rs800292, the genetic deletion of *CFHR3/1*, or (rarely) both. This finding is confirmed by analyses performed using the IAMDGC cohort (see Supplementary Figs. 1, 2, and 4). One protective haplotype, with a frequency of 1.6% and 0.8% among controls and cases of the IAMDGC cohort, respectively, carries a protective allele at rs800292 but not at rs1410996. Conversely, one haplotype containing the minor allele at rs1410996 without protective alleles from rs800292 or the *CFHR3/1* deletion is carried by 0.7% of controls and 0.6% of cases from the IAMDGC cohort and is not associated with protection against AMD ($p = 0.1$, uncorrected for multiple testing). This suggests that the minor allele at rs1410996 is associated with protection solely because it coincides with the protective allele A at rs800292 and the *CFHR3/1* deletion. Causality for protection at the Chr1 AMD locus is therefore more likely to originate from rs800292 or the deletion of *CFHR3/1* than from rs1410996. This observation makes the use of IAMDGC 1.1 to demonstrate overlaps between specific phenotypes, such as the concentration of various proteins in serum and tissue, and AMD-associated variants[42–44,58], problematic. Previous work has shown that IAMDGC 1.1 influences the level of at least 40 serum proteins, with only six of them associated with AMD[58]. It is therefore essential to carefully refine both protein-specific and AMD associations with this variant. In our study, we were for instance able to show that associations between plasma FHR-4 levels and rs1410996 were in fact entirely attributable to rs7531555, a robust *CFHR4* QTL with no independent association with AMD.

We demonstrated that FHR-4 protein was present at the RPE/Bruch's membrane/choroid interface using both immunohistochemistry and ELISA, and that its levels varied with QTLs for liver *CFHR4* expression and protein. The monoclonal antibody used for immunohistochemistry displayed cross-reactivity with recombinant FHR-3 on Western blot but did not detect this protein in normal serum (see Supplementary Fig. 11). Since equimolar concentrations of recombinant proteins were compared in the assay, the cross-reactivity of the antibody with FHR-3 is negligible. The presence of FHR-4 in Bruch's membrane, choroid and drusen from human donor eyes had been previously demonstrated using immunohistochemistry[43]. Altogether, our study indicates that the effect of FHR-4 on AP regulation at this critical interface is limited. FHR-4 has a high level of amino acid homology with SCR domains of FH/FHL-1 but lacks critical regions responsible for co-factor and decay acceleration activity[47]. Previous reports found that FHR-4A may enhance the co-factor activity of FH[48,59] but not FHL-1[48], and that it has no effect on the decay acceleration activity of FH[48]. We did not observe any significant effect of FHR-4 on AMD susceptibility among individuals carrying either the FH/FHL-1 I62 or 402H allotypes, which indicates that the effect of FHR-4 on FH/FHL-1 function at the RPE/Bruch's membrane/choroid is marginal at best. The lack of association between FHR-4 and AMD susceptibility is inconsistent with a competition model for this protein. Competition between FHR-4 and FH/FHL-1 for binding to C3b has so far only been demonstrated for FHR-4B[43], which is below detection limits or not present in serum[46]. Overall, our results demonstrate that *CFHR4* is unlikely to play a significant role in AMD disease mechanisms, and that modulation of FHR-4 has limited therapeutic potential to prevent the initiation of AMD.

Examination of longitudinal clinical data indicates that variation in FHR-4 is unlikely to influence the risk of progression to late AMD or to slow the progression of the disease. It should be noted that the survival analysis that we performed is limited by small sample size and by heterogeneities in AMD severity at first visit between eyes. To address this and to reduce the amount of variability associated with both AMD and FHR-4 variation, we considered individuals homozygous for risk alleles at rs1061170 and with no risk alleles at the Chr10 AMD locus so that cases of AMD were predominantly caused by Chr1-driven complement dysregulation. Restricting our analyses to subjects with homozygosity at rs10494745 (reduced FHR-4) and rs61818956

(elevated FHR-4) ensured that we compared groups of individuals with the largest difference in FHR-4 levels. Given the small number of eyes that met these criteria, additional AMD-associated variants may have influenced our findings. Previous work[60,61] has shown that in addition to the *CFH-CFHR5* and *ARMS2/HTRA1* regions, the *C2-CFB* and *C3* genes may have a significant impact on AMD progression. Our analysis did not allow us to identify or account for the possible effect of confounders associated specifically with progression to neovascular AMD but not cRORA[60] (or vice-versa). Confirmation of our findings using clinical progression data from a larger patient population is therefore warranted.

While our study focuses on FHR-4, it should be noted that two of the lead pQTLs for *CFHR2* identified by previous GWAS[42,44] (rs4085749 and rs3790414) are in complete LD with rs7531555. The rs4085749 variant is also identified in the GTEx v8 dataset as the most strongly associated *CFHR2* splice QTL. The fact that rs7531555 is not independently associated with AMD susceptibility would suggest, analogous to FHR-4, that FHR-2 levels are unlikely to influence AMD susceptibility. Further work is however necessary to confirm this. Efforts to identify serum/plasma proteins associated with AMD have the potential to significantly improve our understanding of AMD disease mechanisms and our ability to discover biomarkers associated with specific disease outcomes[58]. More studies are required to elucidate the precise contributions of systemic AP regulators to the initiation and progression of AMD-associated pathology. Despite being dominant drivers of disease, common risk polymorphisms at the Chr1 AMD locus are not significantly associated with levels of FH/FHL-1 in serum[42,44,58]. Clarifying the role played by systemic AP regulators in AMD requires careful consideration of their effect at the RPE/Bruch's membrane/choroid interface and their potential interactions with the risk and protective forms of FH/FHL-1 and/or FHR-1 and FHR-3. The importance of changes in the structure of the blood-retinal barrier and plasma proteome caused by aging[3] should also not be overlooked.

## Methods

This study complied with all relevant ethical regulations. It adhered to the tenets of the Declaration of Helsinki and was approved by the Institutional Review Boards of the University of Utah, United States.

### Utah & Iowa Study Cohort

Subjects from the Utah & Iowa cohort were recruited between 2009 and 2019 at the Sharon Eccles Steele Center for Translational Medicine, John A. Moran Eye Center, University of Utah, United States, and between 1999 and 2009 at the Cell and Molecular Biology Center, Department of Ophthalmology and Visual Sciences, University of Iowa, Iowa City, Iowa, United States as part of a case/control study of the genetic etiology of AMD. These studies adhered to the tenets of the Declaration of Helsinki, were approved by the Institutional Review Boards of the University of Utah and University of Iowa and did not involve any form of compensation for participants. All subjects provided informed written research consent. Venous blood and/or saliva and demographic data including age, sex, ethnicity, smoking history, and medical history were collected at the time of recruitment. Sex was determined based on self-reporting. Disaggregated sex and gender data were not collected. For each subject, both eyes were graded by the same two independent experienced observers based on fundus photographs and/or spectral domain optical coherence tomography volume scans collected at the time of recruitment. Grading was based on the international classification of mutually exclusive stages of age-related maculopathy introduced by the Rotterdam Group[62]. Patients with no clinically observable signs of AMD (grade 0) formed controls. Patients presenting drusen less than 63 μm in diameter, soft distinct drusen (≥63 μm in diameter) with or without pigmentary changes, isolated pigmentary changes without drusen (≥63 μm in diameter), soft indistinct drusen (≥125 μm in diameter) with or without pigmentary

changes (intermediate AMD), geographic atrophy and/or neovascular AMD (late AMD) in at least one eye were classified as cases[23].

## Genotyping of the Utah & Iowa Cohort and quality control

Genomic DNA was isolated from peripheral blood leukocytes with QIAamp DNA Blood Maxi kits (Qiagen, United States). Genotyping was performed by TaqMan assays (Applied Biosystems, United States) using 10 ng of template DNA in a 5 μL reaction. When available, pre-designed assays were employed. When pre-designed assays were unavailable, custom assays were designed using the manufacturer's software. The thermal cycling conditions in the 384-well thermocycler (PTC-225, MJ Research) consisted of an initial hold at 95 °C for 10 min, followed by 40 cycles of a 15-s 95 °C denaturation step and a 1-min 60 °C annealing and extension step. Plates were read in the 7900HT Fast Real-Time PCR System (Applied Biosystems, United States). Data cleaning and quality control checks were performed using PLINK v1.9[63] [www.cog-genomics.org/plink/1.9/] and PLINK v2.0[64] [http://www.cog-genomics.org/plink/2.0/]. Heterogeneity between subjects recruited in Utah and Iowa were assessed using Cochran's Q-statistic[65] and the $I^2$ metric[66] while adjusting for age and sex. We found no evidence of heterogeneity when considering SNPs and haplotypes in the *CFH-CFHR5* region ($I^2 = 0\%, p > 0.47$). The Utah and Iowa cohorts were therefore combined without resorting to meta-analyses for all associations analyses.

## IAMDGC Cohort

Genetic, demographic, and phenotypic data from the GWAS performed by the International AMD Genomics Consortium (IAMDGC) in 2016[19] was accessed through the database of Genotypes and Phenotypes (dbGaP, Project ID: 30476, PI: Moussa A. Zouache). As per the IAMDGC, relevant data and samples used for genotyping were collected in multiple centers in accordance with principles from the Declaration of Helsinki. All subjects provided informed consent, and protocols were reviewed and approved by local ethics committees. The list of organizations whose ethical committee approved the protocol may be found in the original publication [https://www.nature.com/articles/ng.3448] (Supplementary Table 1). The IAMDGC General Research Use (IAMDGC GRU, dbGaP accession number phs001039.v1.c1) and Disease Specific (IAMDGC DS, dbGaP accession number phs001039.v1.c2) datasets [https://www.ncbi.nlm.nih.gov/projects/gap/cgi-bin/study.cgi?study_id=phs001039.v1.p1], merged using BCFtools[67] [https://github.com/samtools/bcftools], included genotypes, age, sex and AMD severity for 35,358 individuals (males: 15,144 males; females: 20,214). Sex was determined based on self-reporting. Disaggregated sex and gender data were not collected. Subjects presenting late (GA and/or neovascular AMD) or intermediate (pigmentary changes in the RPE or more than five macular drusen greater than 63 μm in diameter) AMD in at least one eye at first diagnosis were classified as cases. IAMDGC controls consisted of subjects with no known advanced or intermediate AMD.

## Genotyping of IAMDGC Cohort, Imputation, and Quality Control

Information on DNA and chip design, annotation, and genotyping platform are detailed in the original publication from the IAMDGC[19]. Cleaning and quality control checks performed by the IAMDGC excluded SNPs with call rates <98.5%, deviation from Hardy−Weinberg equilibrium with $p < 1 \times 10^{-6}$ and mapping to multiple locations. Imputation was performed on the basis of the 1000 Genomes Project phase 1, version 3 (SHAPEIT2 Reference), using Minimac[68]. Altogether, the IAMDGC GRU and DS datasets comprised quality-controlled genotypes for a total of 11,722,963 SNPs (11,294,486 of them imputed). Additional cleaning, quality control checks and variant and subject selection procedures were implemented using PLINK (v2.0)[64]. Allele dosages were used for the imputed data. Rare variants (MAF <1%) and

common variants with poor imputation quality ($r^2 < 0.3$) were filtered out from the combined dataset. This filtering yielded a total of 8,901,744 genotyped (265,543) or imputed (8,636,201) variants.

## Subject selection for genetic analyses

All participants (Utah & Iowa or IAMDGC cohort) younger than 55 at the time of first visit were excluded from analyses. Individuals with genotype call rates <98.5% were also excluded. The IAMDGC filtered-out subjects with discrepancy between reported sex and chromosome information or with atypical sex chromosome configuration as part of their original analysis. For each participant of the IAMDGC study, ancestry information was inferred from genotype data using the first two principal components of autosomal genotyped variants together with genotype information of samples from the Human Genome Diversity Project[69]. Only subjects with self-reported (Utah & Iowa) or genetically determined (IAMDGC) European ancestry were retained. All participants from the Utah & Iowa cohort were unrelated. Since the IAMGC dataset did not include any report or measure of relatedness, KING-robust kinship estimator[70] was used to exclude one member of each pair of subjects that were third-degree relatives or closer (kinship coefficient cutoff of 0.0625). Altogether, this filtering yielded a total of 4787 unrelated European participants (1798 males and 2987 females) in the Utah & Iowa cohort and 30,919 unrelated European subjects (13,182 males and 17,737 females) in the IAMDGC cohort.

## Genetic association analyses

The GRCh37 Homo Sapiens assembly was used for all genetic considerations and analyses. Minor allele frequencies were generated using PLINK (v1.9)[63]. Minor allele and haplotype frequencies among populations with European ancestry were obtained from the 1000 Genomes Project phase 3 (1000 G EUR)[54] [https://www.ncbi.nlm.nih.gov/variation/tools/1000genomes/] either from the LDlink website [https://ldlink.nci.nih.gov/] or by using the R package LDlinkR[71]. Estimates were obtained by combining Utah residents in the United States (CEU), residents from Toscany in Italy (TSI), Finnish individuals in Finland (FIN), British individuals in England and Scotland (GBR) and subjects from the Iberian population in Spain (IBS). Diplotype frequencies among 1000 G EUR individuals were generated by downloading phased genotype data from [https://www.ncbi.nlm.nih.gov/variation/tools/1000genomes/] for all participants and relevant SNPs. Principal components for cases and controls with European ancestry from the IAMDGC cohort were generated using PLINK (v2.0)[64] and high-quality independent ($r^2 > 0.2$) variants. Single-variant association analyses were performed using PLINK (v2.0)[64] using either phased genotypes (Utah & Iowa cohort, phased using the *haplo.stats* package) or allele dosage (IAMDGC cohort). Effect size and *p*-values were adjusted for age and sex for both Utah & Iowa and IAMDGC cohorts, in addition to the first two genetic principal components for the IAMDGC cohort only. Haplotype analyses were performed in R[72] using the package *haplo.stats*[73]. The package uses an energy minimization algorithm to analyze indirectly measured haplotypes and assumes that all subjects are unrelated. The *haplo.glm* function was used to perform generalized regressions of AMD status on haplotype effects while adjusting for age, sex for both Utah & Iowa and IAMDGC cohorts, in addition to the first two genetic principal components for the IAMDGC cohort. The function uses the posterior probabilities of pairs of haplotypes per subjects as weights to update regression coefficients. The *haplo.stats* package was used to infer most likely haplotypes among subjects from the Utah & Iowa and IAMDGC cohorts. Combinations of most likely haplotypes were used to infer diplotypes for each participant. Logistic regression models (*glm* function in R) adjusted for age and sex for both Utah & Iowa and IAMDGC cohorts, in addition to the first two genetic principal components for the IAMDGC cohort, were applied to generate effect sizes and *p*-values. In addition to logistic regressions, associations between AMD and single variants,

haplotypes, or diplotypes were also assessed using the $\chi^2$ test for association. For all analyses, multiple testing was accounted for by adjusting the significance level using Bonferroni corrections. Manhattan plots were plotted using the R package CMplot[74]. When considering multiple variants, the most parsimonious best-fit regression was determined on the basis of likelihood ratio test statistics.

## Identification of independent CFHR4 cis-eQTLs

The Genotype-Tissue Expression Project (GTEx)[54] dataset was accessed through the database of Genotypes and Phenotypes (dbGaP, accession number phs000424.v8.p2.c1, Project ID: 28906, PI: Moussa A. Zouache) [https://www.ncbi.nlm.nih.gov/projects/gap/cgi-bin/study.cgi?study_id=phs000424.v8.p2]. Methods applied by the GTEx consortium for collection, preprocessing and exclusion of samples, quality control checks, and gene expression quantification and normalization are detailed online at [https://gtexportal.org/home/methods]. Additional cleaning, quality control checks, variant and subject filtering procedures, and identifications of variants in linkage disequilibrium were implemented using PLINK (v2.0)[64]. Expression data specific to the liver was available for 226 donors. Twenty-six donors categorized as non-white (5 Asians, 20 African Americans, 1 with unknown ancestry) by the GTEx consortium were filtered out of the dataset, yielding 200 donors. Out of these, 183 had genotype information (whole genome sequencing) available (129 males, 54 females, median age 56 IQR 14.5). SNPs with a call rate inferior to 0.1 and variants with a minor allele frequency lower than 5% were excluded from analyses. Analyses of expression quantitative trait loci (eQTL) were performed using QTLtools v1.3.1[75] [https://qtltools.github.io/qtltools/] and R[72]. Associations between gene expression levels and SNPs in cis were assessed through linear regression models adjusted for covariates, generated by the GTEx consortium, consisting of the first 5 genotyping principal components, a set of 30 covariates identified using the Probabilistic Estimation of Expression Residuals (PEER) method[76], sequencing platform, sequencing protocol, and sex. A forward-backward stepwise regression was implemented to determine the number of independent cis-eQTLs associated with CFHR4 expression levels and to assign all significant variants to the independent signal they relate to reference [75]. A permutation pass involving 5000 permutations and rank normal transformation of CFHR4 expression levels was run on chromosome 1 between 150,000,000–200,000,000 bp. Based on this permutation pass, the nominal p-value threshold for a false discovery rate of 0.1 used for the conditional analysis to identify independent eQTLs was set at 2.0e-05. Independent signals were selected based on the backward stepwise regression pass, which incorporated all significant signals.

## Identification of independent CFHR4 cis-pQTLs

Independent serum or plasma protein expression quantitative loci (pQTL) for CFHR4, listed in Supplementary Table S5, were identified by mining results from two large blood plasma genome-proteome-wide association studies. The first one, published by Pietzner et al. (2021)[55], included 4775 protein targets measured in plasma from 10,708 individuals with European ancestry (mean age 48.6 years, 53.3% women) from the Finland study. The list of pQTLs for CFHR4, which included for each variant the effect allele (EA) with associated frequency, an estimated effect size (adjusted for age, sex, the first ten genetic principal components and test site), p-value (stringent Bonferroni threshold for significance: p < 1.004e-11), and direction of association was downloaded from [https://omicscience.org/apps/pgwas/pgwas.download.php]. Outputs of the conditional analysis performed by the authors, which was implemented in GCTA[77] using a collinear cutoff of 0.1 and a p-value threshold of 1.004e-11, were obtained from the original publication. The second blood plasma genome-proteome-wide association study considered, published by Gudjonsson et al.[56], included 4,782 protein

targets measured in serum from 5,368 individuals with European ancestry. Outcomes from the association analyses included for each variant the EA and associated frequency, an estimated effect size (adjusted for age, sex, five genetic principal components and genotyping platforms) and p-value (Bonferroni threshold for significance: p < 1.046e-11). The stepwise conditional association analysis, implemented in GCTA-COJO[77,78] to identify independent pQTLs, conditioned on variants with the lowest p-value while keeping track of variants that reached study-wide significant associations. The outcome from this analysis was obtained from the original publication. Further details regarding the methodology implemented by each study may be retrieved from the original publications. Only pQTLs with frequency larger than 5% and with p-value for single-variant association inferior to the threshold set by each study were retained.

## Human plasma sample selection and processing

Plasma samples were collected at the time of enrollment for a total of 4291 subjects from the Utah & Iowa Cohort and stored at −80 °C. All samples selected for quantification of FHR-4 levels were from individuals with self-reported European ancestry and between the age of 45–95. All of them had been screened for AMD (no AMD, intermediate AMD, or late AMD) in both eyes at the time of sample collection using multimodal imaging (fundus photographs, infrared and optical coherence tomography volume scans). To ensure that AMD susceptibility in these subjects was primarily determined by the Chr1 AMD locus, only subjects with no risk alleles at the Chr10 AMD locus were included. Samples were selected based on homozygous diplotype combinations (Fig. 1D) and genotypes at variants identified as genetic modulators of FHR-4 levels, including genetic recombinants. Whenever possible, subjects were also age-matched.

## Human tissue sample selection and processing

Human donor eye tissues were selected from a repository composed of more than 9000 pairs of human eyes processed, flash-frozen, or fixed within five hours of death held at the Sharon Eccles Steele Center for Translational Medicine, John A. Moran Eye Center, University of Utah. Fresh human donor eyes were processed to isolate the RPE/Bruch's membrane/choroid tissue layer, collected from the nasal part of each eye in a region approximately 2–5 mm from the optic nerve head, flash-frozen in liquid nitrogen and stored in a −80 °C freezer. Vitreous humor was recovered from each eye at the time of donor eye collection, flash-frozen, and stored at −80 °C. For total protein extraction, RPE/Bruch's membrane/choroid tissues were thawed on ice and then washed once with 300 μL cold 1X PBS containing 1% Halt protease and phosphatase inhibitor cocktail with EDTA (Cat. #78440, ThermoFisher Scientific, United States). After a single wash, tissues were resuspended in 300 μl Tissue Protein Extraction Reagent (T-PER, Cat. #78510, ThermoFisher Scientific, United States) containing 1% Halt protease and phosphatase inhibitor cocktail with EDTA. Samples were then homogenized on ice using a probe sonicator until the pellet was broken into small pieces. This step was followed by shaking at $1500 \times g$ every 20 s at 4 °C overnight. Portions of vitreous humor were partially thawed on ice before being weighed and adding an equal volume of T-PER solution containing 1% Halt protease and phosphatase inhibitor cocktail with EDTA. The mix was sonicated and then processed identically as described for RPE/Bruch's membrane/choroid tissue samples. T-PER processed material was centrifuged at $24,000 \times g$ for 5 min and supernatant transferred to a 96-deep well storage plates. Plasma, RPE/Bruch's membrane/choroid and vitreous protein lysates were then aliquoted from deep well plates into multiple daughter plates for total and FHR-4 protein assessment. To determine total protein concentration a 660 nm protein assay kit (Pierce 660 nm Protein Assay Kit, Cat. #22662, ThermoFisher Scientific, United States) was used according to the manufacturer's protocol.

## Factor H-related 4 protein measurements in plasma, serum and tissue samples

The level of FHR-4 protein in plasma, serum, soluble RPE/Bruch's membrane/choroid, and vitreous humor lysates was measured using a custom sandwich ELISA. Black MaxiSorp 96-well microplate wells were coated with 100 μL of anti-FHR-4 monoclonal mouse antibody (Cat. #MAB5980, R&D Systems, United States) diluted to 1:1000 in 50 mM carbonate coating buffer (pH 9.6). After covering and incubating plates overnight at 4 °C, wells were washed using a BioTek EL404 Microplate washer (Agilent Technologies, United States) consisting of 3 cycles of 300 μl PBST (1X PBS + 0.05% Tween 20) with 5 s of soak time and no shaking followed by aspiration, blocked for 1 h in 1% bovine serum albumin (BSA) in 1X PBS, and washed as above. One-hundred microliters of plasma or ocular protein lysate were diluted 1:2500 or 1:25, respectively in reagent dilution buffer (RDB; 1X PBS + 1% Bovine Serum Albumin Fraction V), added to wells in duplicate, and incubated for 1.5 h at room temperature. The plate was washed as above and then incubated for 1.5 h with a polyclonal sheep anti-FHR-4 antibody (Cat. #AF5980, R&D Systems, United States) diluted to 1:1000. The plate was washed again and then incubated for 1 h with a horse-radish peroxidase-labeled polyclonal rabbit anti-sheep antibody (Cat. #313-035-045, Jackson ImmunoResearch, United States) diluted at 1:10,000. The plate was washed as above, before being incubated for 5 min with SuperSignal ELISA pico chemiluminescent substrate (Cat. #37069, ThermoFisher Scientific, United States). Luminescence was measured using a BioTek Synergy4 plate reader (Agilent Technologies, United States). Protein concentrations were interpolated from recombinant C-terminal 6x-His-tagged FHR-4A protein (produced in standard mammalian cell lines by vendor Yurogen Inc, United States) standard curve using Biotek Synergy software and non-linear 4 parameter curve analysis (Agilent Technologies, United States). Plasma or serum samples were normalized by volume (ng/mL) and both RPE/Bruch's membrane/choroid and vitreous humor samples were normalized to total protein lysate concentration (ng/mg). The specificity of the monoclonal mouse anti-FHR-4 antibody (Cat. MAB5980, R&D systems) and polyclonal sheep anti-FHR-4 antibody (Cat. #AF5980, R&D Systems, United States) to FHR-4A and FHR-4B and cross-reactivity to complement factor H family of proteins tested with recombinants of the complement factor H family of proteins factor H (FH), factor H-like 1 (FHL-1), factor H-related 1 A (FHR-1A), factor H-related 1B (FHR-1B), factor H-related 2 (FHR-2), factor H-related 3 (FHR-3) and factor H-related 5 (FHR-5) was determined by Western blot (see Supplementary Fig. 11).

## Fit-for-purpose and validation of FHR-4 ELISA

To determine the robustness of the FHR-4 ELISA we performed a fit-for-purpose study to establish recombinant FHR-4A protein standard curve intra-assay, inter-assay, and dilutional linearity. The intra-assay and inter-assay experiment were performed over multiple days in multiple replicate wells using reagent dilution buffer as the matrix. Dilutional linearity was similarly performed by spiking 100 ng/mL recombinant FHR-4A protein into FH-depleted serum (Cat. #A337, Complement Technology, United States) diluted 1:1000 and subsequently diluted 2-fold in non-spiked 1:1000 FH-depleted serum. Parallelism studies used normal human serum complement standard (Cat. #A100, Lot # 116207, Quidel, United States) or pools of normal human plasma samples (isolated and pooled in-house from 4 subjects). To determine FHR-4 ELISA cross-reactivity, recombinant his-tagged FH, FHL-1, FHR-1A, and FHR-3 proteins were purified in-house or purchased from commercial vendors: FHR-2 (Cat. #91-468, ProSci, United States), FHR-4A (Yurogen Inc, United States), FHR-4B (Cat. #5980-CH-050, R&D Systems, United States), FHR-5 (Cat. #3845-F5-050 R&D Systems, United States) and spiked into FH-depleted serum (1:1000 dilution) at equimolar concentrations ranging from 0.001 to 10 nM. To confirm FHR-4 specificity and selectivity in human plasma (Caucasian cohort)

and serum (Rapa Nui cohort), subjects with 0, 1, or 2 copies of the CFHR1/4 genetic deletion were analyzed in the FHR-4 ELISA at 1:2500 dilution; see Supplementary Fig. 5. The number of copies of the CFHR1/4 deletion among individuals from the Utah & Iowa and Rapa Nui cohorts was determined through copy-number variation reactions.

## Statistical analyses of associations with FHR-4 levels

Associations with FHR-4 levels were assessed using R[72]. Measured FHR-4 levels were natural logarithmically transformed and scaled. The Kruskal–Wallis test was used to test associations between FHR-4-levels and categorical variables including sex, AMD status (no AMD or AMD), AMD severity (no AMD, early/intermediate AMD or late AMD) and CFH-CFHR5 diplotype. When significant associations were identified, post-hoc pairwise comparisons were performed using the Conover–Iman test implemented in the conover.test package[79]. All p-values were two-sided and adjusted for multiple testing using the Bonferroni correction. A linear regression of FHR-4 levels against age was used to determine the association between protein levels and increasing age. For the fit-for-purpose study of our in-house FHR-4 ELISA, the Mann-Whitney t-test was used to assess differences in FHR-4 levels among subjects with 0, 1, or 2 CFHR1/4 deletions. Power analyses performed using the R[72] package pwr indicated that the number of plasma samples considered provided a power higher than 80% to detect associations between FHR-4 levels and QTLs in plasma and tissue.

## Copy-number variation reaction assay for CFHR1/CFHR4

Total genomic DNA was extracted from whole blood using the QIAamp DNA Blood Maxi Kit. CFHR1/CFHR4 copy number was determined using Taqman (Applied Biosystems [ABI], United States) under the manufacturer's recommended conditions. Pre-designed copy-number assays for CFHR1 (Hs04197581) and CFHR4 (Hs03350272) were used, as well as an additional CFHR1-CFHR4 intergenic assay (Hs04214423). Samples were amplified and analyzed using the 7900HT Fast Real-Time PCR System (Applied Biosystems, United States). Thermal cycling consisted of an initial hold at 95 °C for 10 min, followed by 40 cycles of a 15-s 95 °C denaturation step and a 1-min 60 °C annealing and extension step. Each sample was amplified in duplicate. An internal single-copy reference assay, RNAse P, was included in every reaction. Copy number was estimated using the CopyCaller software (Applied Biosystems, United States).

## FHR-4 immunohistochemistry

Human donor eyes used for immunohistochemistry were fixed in 4% paraformaldehyde and stored at 4 °C. Glass slides containing paraffin-embedded tissue sections were deparaffinized in Slidebrite (Cat. #23401, Electron Microscopy Sciences, United States) and rehydrated in graded ethanol baths. Antigen retrieval was performed by incubating the slides in citrate buffer (Cat. #C9999, Sigma-Aldrich, United States) for 30 min at 95 °C using a decloaking chamber (Cat. #DC2012, Biocare, United States). Slides were rinsed in 20 mM Tris-buffered saline pH 7.4 (1X TBS) and incubated with Bloxall (Cat. #SP-6000, Vector Laboratories, United States) for 10 min. Non-specific antigens were blocked by incubating slides for 1 h in 1X TBS containing 1% bovine serum albumin (BSA) and 2% normal horse serum. Slides were incubated with the mouse monoclonal anti-FHR-4 antibody (Cat. #MAB5980, R&D Systems, United States) diluted 1:200 in 1X TBS containing 1% BSA overnight at 4 °C. After rinsing in TBS, the slides were incubated with ImmPRESS-Alkaline Phosphatase horse anti-mouse IgG polymer reagent (Cat. #MP-5402, Vector Laboratories, United States) for 30 min at room temperature. A working solution of ImmPACT Vector Red substrate (Cat. #SK-5105, Vector Laboratories, United States) was prepared as directed by the manufacturer and added to the slides for 5 min. After rinsing in 1X TBS and deionized water, the slides were counterstained with a 1:10 dilution of Mayer's Hematoxylin (Cat. #72804, ThermoFisher Scientific, United States) for

1 min. Slides were dehydrated by incubating in a series of graded ethanol baths for 5 min per wash and coverslips were mounted using DEPEX mounting medium (Cat. #13515, Electron Microscopy Sciences, United States). An image of a negative control may be found in Supplementary Fig. 12. The specificity of the anti-FHR-4 antibody and cross-reactivity to complement factor H family of proteins was tested by Western blot. Equimolar concentrations of recombinant protein were used for this assay. The antibody displayed a cross-reactivity with recombinant FHR-3 (see Supplementary Fig. 11). A band consistent with FHR-3 in normal serum was not detected. No cross-reactivity with any of the other recombinants was observed.

### Disease Progression Among Genetically Stratified AMD Patients

Analyses of progression data were performed for patients from the Utah cohort with a Risk/Risk diplotype at the Chr1 AMD locus and no risk alleles at *ARMS2/HTRA1* and presenting clinical signs of AMD. Refined grading of images and the classification of AMD used for these analyses were previously described[16]. Briefly, eyes were graded as having early AMD (here labeled as AMD grade 2) if drusen with diameters between 63 μm and 125 μm and no hyperpigmentary changes were detected on optical coherence tomography B-scans. Eyes with drusen larger than 125 μm in diameter or exhibiting hyperpigmentary changes were classified as presenting intermediate AMD (AMD grade 3). If large pigment epithelial detachments (dome-shaped elevations >1000 μm in basal diameter and >100 μm in maximum heights) were present, these eyes were given an AMD grade 4. We included the recently introduced AMD grade of incomplete RPE and outer retinal atrophy (iRORA) as the intermediate AMD grade 5[80,81]. Classification of late-stage exudative neovascular AMD was based on clinical documentation (*i.e.*, history of intravitreal anti-vascular endothelial growth factor injection and corresponding signs on imaging data). The date of conversion was defined as the earliest documented exudation due to neovascular AMD or complete RPE and outer retinal atrophy (cRORA)[80] associated with late-stage atrophic AMD in an eye with a previous diagnosis of early or intermediate AMD. The presence and conversion to late-stage neovascular AMD were confirmed by reviewing all available imaging data and based on the presence of disease activity signs, including intraretinal or subretinal fluid or hemorrhage. Eyes for which the time between first visit with AMD and date of initial conversion could not be determined were excluded from analyses.

### Survival analyses

The method applied to perform survival analyses of risk of progression to late-stage AMD have been described previously[16]. Briefly, we defined for each eye the first visit as the earliest visit available with imaging-based documentation of a clinical phenotype of AMD. The timepoint of conversion to late AMD was defined as the first occurrence of either late-stage atrophic or late-stage exudative AMD. Conversion to atrophic and neovascular AMD were also evaluated separately. For all analyses, eyes were censored if no conversion had occurred by the last recorded visit. Further, eyes were censored at the timepoint of any potential confounding event. These included occurrence of other retinal diseases (*e.g.*, branch retinal vein occlusion) or retinal surgery (*e.g.* retinal detachment surgery). Survival analyses were performed using R[72] and packages *survival*[82], *survminer*, and *coxme*[83]. Cox proportional hazard (CoxPH) models were applied separately to both eye- and patient-level analyses. Eye-level analyses used only one eye. The eye selected was either (1) the only eye that converted; (2) the eye that converted first if both eyes were at risk; and (3) the eye with the more severe AMD grade at the first visit if both eyes converted at the same timepoint. Patient-level models used both eyes of each subject and included a frailty term to account for correlations between these eyes. Log-likelihood statistics were used to determine if covariates including gender, age and AMD grade assessed at the first visit should be included. The log-rank test was employed to compare survival distributions. The significance level for all statistical tests was set at α = 0.05. All *p*-values generated are two-sided. When necessary, *p*-values were adjusted for multiple testing using a Bonferroni correction.

### Reporting summary

Further information on research design is available in the Nature Portfolio Reporting Summary linked to this article.

## Data availability

Genotypes and gene expression data from The Genotype-Tissue Expression Project (GTEx) v8[54], associated demographic and tissue information (including age and ethnicity) and covariates used in eQTL analyses may be obtained either through the database of Genotypes and Phenotypes (dbGAP) (accession number phs000424.v8.p2.c1) [https://www.ncbi.nlm.nih.gov/projects/gap/cgi-bin/study.cgi?study_id=phs000424.v8.p2] or through the GTex dataset page [https://gtexportal.org/home/datasets]. Genotypes, case/control status and demographic information including age, gender and ethnicity from The IAMDGC GWAS[19] may be accessed through dbGaP (accession numbers phs001039.v1.c1 and phs001039.v1.c2) [https://www.ncbi.nlm.nih.gov/projects/gap/cgi-bin/study.cgi?study_id=phs001039.v1.p1]. The list of pQTLs for *CFHR4* published by Pietzner et al. (2021)[55] (based on 10,708 subjects with European ancestry) may be downloaded from [https://omicscience.org/apps/pgwas/pgwas.download.php]. Outputs of the conditional analysis performed by the group may be downloaded from the original publication ([https://www.science.org/doi/10.1126/science.abj1541], Supplementary Table S2). The list of pQTLs for *CFHR4* published by Gudjonsson et al. (2022)[56] may be downloaded from the original publication [https://www.nature.com/articles/s41467-021-27850-z]. Outputs of the conditional analysis were obtained from the original publication (Supplementary Data 3). All summary information pertaining to the Utah & Iowa cohorts or human donor eyes is included in the manuscript. Summary statistics of associations are also provided in the manuscript or its Supplementary Information. Raw data for FHR-4 levels in ocular tissue (RPE/Bruch's Membrane/choroid) and associated genotype at *CFHR4* QTLs for each donor are provided in the Source Data file. Raw data for FHR-4 levels in plasma/serum and associated genotype at *CFHR4* QTLs for each subject are provided in the Source Data file. Genotypes for AMD-associated variants and clinical information (used for survival analyses) for the Utah & Iowa cohort are not available due to data privacy (sharing or submission of these information to external repositories or databases is limited by our IRB). We are however open to collaborative work seeking to harness these resources. Source data are provided with this paper.

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

## Acknowledgements

This work was supported in part by the National Eye Institute of the National Institutes of Health under award numbers R24EY017404 (GSH), philanthropic donations to the Sharon Eccles Steele Center for Translational Medicine, John A. Moran Eye Center, University of Utah, USA, and an Unrestricted Grant from Research to Prevent Blindness, New York, NY, to the Department of Ophthalmology & Visual Sciences, University of Utah, USA.

## Author contributions

M.A.Z. and B.T.R. contributed equally to the work as first authors. M.A.Z. wrote the manuscript and generated all figures. M.A.Z., B.T.R., C.M.P., B.L.W., and G.S.H. reviewed the manuscript. M.A.Z., B.T.R., and C.M.P. designed all protocols for the study. M.A.Z., B.T.R., C.M.P., and G.S.H. supervised the study. G.S.H. provided financial support for the research. Imputation, quality control, filtering, and phasing of genotype data was implemented by M.A.Z. Refinement of genetic associations with AMD were performed by M.A.Z. (IAMGC and Utah & Iowa cohorts) and C.M.P. (Utah & Iowa cohort). Identification of independent eQTLs and pQTLs was performed by M.A.Z., with contribution from C.M.P. and B.T.R. W.C.H., J.T., and J.L.H. managed recruitment of AMD cases and controls for the Utah & Iowa cohorts, collected blood samples, relevant demographic information and clinical images used for grading. G.S.H. and J.L.H. determined the AMD status of all cases and controls. S.S.V. and M.F. provided refined assessment of AMD severity for patients included in the study. C.M.P. and S.M. processed and genotyped all samples from the Utah & Iowa Cohort and performed quality controls. M.A.Z. performed all analyses of AMD progression (survival analyses). R.A.A., J.L., and T.C. designed, validated, and implemented the ELISA for FHR-4 under the direction and supervision of B.T.R. M.A.Z. and B.T.R. performed statistical analyses of FHR-4 levels in serum and tissue. C.M.P. and S.M. managed the repository of human donor eyes. N.A.S. and B.L.W. performed immunohistochemistry of human tissue. M.A.Z., B.T.R., and C.M.P. all contributed significantly to the work through scientific discussions and critical assessment of data. All authors read and approved the final manuscript.

## Competing interests

G.S.H. is a shareholder, consultant, and co-founder of Perceive Biotherapeutics, LLC. G.S.H., C.M.P., B.L.W., and B.T.R. are inventors on patents and patent applications owned by the University of Utah. J.L.H. is the spouse of G.S.H. None of the other authors have any relevant proprietary interests.
