## [Peer Review File · Nature Communications]

Levels of Complement Factor H-related 4 Protein do Not Influence Susceptibility to Age-related Macular Degeneration or its Course of ProgressionReviewers' Comments:

Reviewer #1:

Remarks to the Author:

The work by Zouache et al. comprehensively addresses the contribution of genetic variants to AMD susceptibility by studying very large cohorts of patients and controls. In addition, they show that variations in FHR-4 levels are entirely explained by three independent quantitative trait loci (QTL). Importantly, the authors show that these FHR-4 QTLs are not independently associated with AMD, and that systemic FHR-4 levels are not independently influenced by AMD-associated haplotypes. Moreover, Zouache et al. show that FHR-4 levels do not influence disease susceptibility or progression and, hence, FHR-4 is unlikely to play an independent role in AMD disease initiation or progression. The paper is very pertinent and addresses the very complicated issue of dissecting the role of genetic variants in the FH gene family, as there is a high linkage disequilibrium between them. For instance, it was previously reported that FHR-4 play a key role in AMD pathogenesis. However, this work convincingly demonstrated the contrary. The manuscript is very well done, but I believe that it is too dense and difficult to follow sometimes and could be simplified to make it easier to understand.

Major comments

- Page 6. Results section "Subject and donor stratification based on AMD susceptibility". The description of the genetic variants at loci 1.1, 1.6 and 1.5 is very confusing after the previous paragraph where it is mentioned that only 3 SNPs are sufficient to explain susceptibility to AMD. The authors should introduce that paragraph in some way to facilitate understanding, perhaps by indicating that other loci are also associated but do not actually independently contribute to the susceptibility for the reasons described below.
- I am not sure what is the added value of analysing cis-eQTL in the result section "Genetic determinants of FHR-4 levels in plasma". In fact, the cis-eQTLs identified are not used in subsequent analysis. Can this part be removed?
- Although the ELISA method used to quantify plasma FHR-4 levels seems to be very specific of FHR-4, the authors should also show the specificity of the antibodies in western blot assays using plasma samples. Particularly in the case of the monoclonal anti-FHR-4 antibody (Cat. MAB5980, R&D systems), as is the one used in the immunohistochemistry studies, it is important to show that there is no cross-reactivity with other FHR proteins.
- The sentence "Immunohistochemistry of human donor eyes confirmed that FHR-4 was present in Bruch's membrane and between pillars of the choriocapillaris (see Fig. 2d)." at the end of the results section "Confirmation of QTL effects in plasma samples", does not make any sense in that part of the manuscript. Perhaps, it can be relocated in section "Liver QTLs influence FHR-4 levels in RPE/Bruch's membrane/choroid and vitreous samples".
- It has been previously described that FHR-4 plasma levels are elevated in AMD patients compared to controls (Cipriani et al. Nature Communications, 2020, 11:778, <https://doi.org/10.1038/s41467-020-14499-3>). Why do you think you have discrepant results?
- Why should "liver" be included in the titles "Liver QTLs influence FHR-4 levels in RPE/Bruch's membrane..." and "Liver QTLs are not independently associated with AMD"? I understand that FHR-4 is mainly produced by the liver, but are the authors excluding the effect of these QTLs in any other cell type that may express the protein?
- The section of the results "Liver QTLs are not independently associated with AMD" seems redundant to me, as this part could be deduced from the first section of the results, where the 3 SNPs that determine susceptibility to the disease are described. Also, I am not sure what the rationale is for result section "Variation in FHR-4 plasma levels does not influence AMD risk or protection", since the authors described that FHR-4 plasma levels are not significantly different between controls, early/intermediate AMD, and late AMD samples (Suppl. Figure S5). Therefore, I suggest deleting some of the sections or, at least, simplifying them.

Minor comments

- To mention the Figures and Tables in order of appearance in the manuscript. For instance, figure 1B should be fig.1A and vice versa, as Fig. 1B is mentioned before (introduction). The same situation applies to supplemental tables and figures.
- Figure 2A. I suggest including the schematic of the genes where the SNPs are located at the bottom of the graph (similar to fig. 1A), as it will serve as a guide for locating the variants.
- Suppl. Figure 10C. To change the colour of the black circles to depict samples with 2 copies in the graph, so the SD bars can be seen.

Reviewer #2:

Remarks to the Author:

The 3 page introduction is a review of AMD genetics and references are mostly the authors' papers. This could be shortened considerably and could include references from other groups.

Figure 4- small numbers. What was the effect of all other genetic variants given that this group has a specific risk variant among many others that could also influence risk. Several cells in this figure have fewer than 5 individuals. What were results without making these subgroups?

A paper relevant to the topic was not mentioned or addressed.

Raychaudhuri S, Ripke S, Li M, Neale BM, et al. Associations of CFHR1-CFHR3 deletion and a CFH SNP to age-related macular degeneration are not independent. *Nat Gen* 2010; 42:553-555.

Much effort was made to determine which CFH snp among a few in high LD were related to AMD, discounting one of them based on results in 0.6 to 0.7% of the cases and controls.

Whereas the main goal was to show that previously published CFHR-4 results were not accurate.

Results require independent confirmation.

Reviewer #3:

Remarks to the Author:

It was a tremendous pleasure reading this manuscript by Dr Zouache and co-authors reporting that variation in FHR4 levels in the human body are not significantly associated with susceptibility to AMD as well as disease progression. The analysis here appears to be vigorously, critically, and transparently done. Here, the Authors arrived at the conclusion that variation in FHR-4 is unlikely to be related to AMD risk because both elevated and reduced plasma and ocular tissue levels of FHR-4 coincide with a higher risk for AMD. I have several very humble suggestions for the Authors:

1. The key conclusions of this manuscript is not consistent with that by Cipriani et al., 2020 (Reference #43, main text). There is thus a robust, collegial and constructive scientific discussion to be had that will strongly benefit readers and the AMD research community. The key point up for discussion would be the findings in this submission that "Variation in FHR-4 Plasma Levels does Not Influence AMD Risk or Protection" whereas Cipriani et al., observed that "Systemic FHR-4 levels are elevated in advanced AMD cases". I would encourage the Authors here to discuss their present findings side by side with Cipriani et al; as one key observable difference was that whereas Cipriani et al., evaluated 484 late AMD patients and 522 phenotyped controls, the present submission evaluates samples in 486 individuals from the Utah & Iowa cohort (median age 75.0 IQR 13.0) with no AMD (235 subjects), early/intermediate AMD (120 subjects) or late AMD (131 subjects) (lines 248 to 250, main text).

2. The Authors observed that single-variant association analyses performed using the Utah & Iowa cohort and validated against the IAMDGC cohort indicated that rs10494745 and rs61818956, which have opposing effects on FHR-4 levels, were both associated with an increased risk for AMD. Could they explain whether the opposing effect of both key variants on FHR-4 levels were solely responsible for them not observing an association between FHR-4 levels and disease risk? This is a key question that should be clarified for the reader because Cipriani et al., also presented their analysis based on IAMDGC SNPs on the chromosome 1 locus. Perhaps, a vigorous analysis on 'where they differ between the 2 papers' would be helpful, as both studies enrolled participants of predominantly European ancestry, and the genetic architecture, haplotypic variation and haplotypic relationships should not differ by much between the 2 studies.

3. In its present form, the manuscript is written with many rs-ids and terminologies (for example

H7, line 313 main text; H5 line 310) describing the extremely complicated relationship between FHR-4 levels, genetic variants (and haplotypes) associated with AMD risk, as well as genetic variants (and haplotypes) associated with FHR-4 levels (QTLs). Even for a self-confessed AMD-enthusiast such as myself, I had to struggle through the rs-ids between text and figures multiple times to reconcile them. Could the Authors consider simplifying the presentation to make it easier for the reader to appreciate the question and the science behind it?

Levels of Complement Factor H-related 4 Protein do Not Influence Susceptibility to Age-related Macular Degeneration or its Course of Progression (Zouache *et al.*) - Response to Reviewers

We would like to thank all reviewers for the time spent reviewing our manuscript and for their helpful comments and suggestions.

Reviewer 1

The work by Zouache *et al.* comprehensively addresses the contribution of genetic variants to AMD susceptibility by studying very large cohorts of patients and controls. In addition, they show that variations in FHR-4 levels are entirely explained by three independent quantitative trait loci (QTL). Importantly, the authors show that these FHR-4 QTLs are not independently associated with AMD, and that systemic FHR-4 levels are not independently influenced by AMD-associated haplotypes. Moreover, Zouache *et al.* show that FHR-4 levels do not influence disease susceptibility or progression and, hence, FHR-4 is unlikely to play an independent role in AMD disease initiation or progression. The paper is very pertinent and addresses the very complicated issue of dissecting the role of genetic variants in the FH gene family, as there is a high linkage disequilibrium between them. For instance, it was previously reported that FHR-4 play a key role in AMD pathogenesis. However, this work convincingly demonstrated the contrary. The manuscript is very well done, but I believe that it is too dense and difficult to follow sometimes and could be simplified to make it easier to understand.

Major comments

1. Page 6. Results section “Subject and donor stratification based on AMD susceptibility”. The description of the genetic variants at loci 1.1, 1.6 and 1.5 is very confusing after the previous paragraph where it is mentioned that only 3 SNPs are sufficient to explain susceptibility to AMD. The authors should introduce that paragraph in some way to facilitate understanding, perhaps by indicating that other loci are also associated but do not actually independently contribute to the susceptibility for the reasons described below.

We modified this paragraph to clarify the refinement of genetic associations within the *CFH-CFHR5* region. We now list all variants independently associated with AMD within this region and proceed to explain that all of these associations can be traced to the three SNPs rs1061170, rs800292 and rs12144939. The revised paragraph is:

*Eight credible sets of variants have been independently associated with AMD within the extended *CFH-CFHR5* region through GWAS¹⁹. Out of the eight index SNPs (IAMDGC Locus 1.1 to 1.8) describing these sets, four (IAMDGC Loci 1.3, 1.4, 1.7 and 1.8) are rare (frequency < 1%) and were therefore excluded from further analyses. IAMDGC Locus 1.2 is in perfect linkage equilibrium with the risk-associated rs1061170 (*CFH* Y402H, OR_{IAMDGC} : 2.18 [2.10; 2.26], $p = 1.1 \times 10^{-422}$); see Supplementary Table 2. The non-coding IAMDGC Locus 1.1 (OR_{IAMDGC} : 0.44 [0.42; 0.46], $p = 4.5 \times 10^{-415}$), tagged by rs10922109 and rs1410996 ($r^2=0.9919$, $D' = 0.9959$) is generally used as a proxy for the combination of haplotypes containing the rs800292 variant (*CFH* I62V, OR_{IAMDGC} : 0.51 [0.49; 0.53], $p = 4.6 \times 10^{-202}$) or the *CFHR3/1* deletion (tagged by rs12144939, OR_{IAMDGC} : 0.50 [0.47; 0.52], $p = 7.1 \times 10^{-187}$). However, this variant does not tag perfectly these two forms of genetic protection. Approximately 1.9% of controls and 1% of cases in the Utah & Iowa cohort and 1.6% of controls and 0.8% of cases in the IAMDGC cohort have a haplotype associated with protection against AMD that carries the protective A allele at rs800292 in the absence of the minor allele at IAMDGC Locus 1.1 (see Supplementary Figures 1 and 2). We therefore use the combination of rs800292 and rs12144939 in place of rs1410996. This choice also ensures that the origin of the genetic protection among individuals protected against AMD can be readily identified. IAMDGC Locus 1.6 (rs6181825, OR_{IAMDGC} : 0.63 [0.60; 0.65], $p = 2.1 \times 10^{-}$*

¹³¹) is present on haplotypes associated with risk, protection or neutrality (defined as the absence of risk or protection²³) for AMD (see Supplementary Figure 3). In addition, the significance of its association with AMD is lost when conditioning regression models on rs1061170, rs800292 and rs12144939 (see Supplementary Figures 2 and 4). Therefore, this variant does not differentiate disease susceptibility and was excluded from further consideration. Finally, the risk allele for IAMDGC Locus 1.5 (rs187328863, OR_{IAMDGC} : 2.30 [2.08; 2.55], $p = 1.1 \times 10^{-57}$) resides exclusively on a less common haplotype (present in 5% of cases and 2.6% of controls in the Utah & Iowa cohort and in 3.5% of cases and 1.7% of controls in the IAMDGC cohort) containing the risk-associated C allele at rs1061170. Its contribution to AMD susceptibility is therefore accounted for by haplotypes including a risk allele at rs1061170. Altogether, these analyses indicate that three SNPs, consisting of the risk-associated rs1061170, the protection-associated rs800292 and rs12144939, a SNP tagging the CFHR3/1 deletion haplotype, are the most likely causal AMD variants within the CFH-CFHR5 extended region and are sufficient to describe the spectrum of common AMD susceptibility associated with the Chr1 AMD locus among individuals with European ancestry (see Fig. 1b and Supplementary Figures 2 and 4).

2. I am not sure what is the added value of analyzing *cis*-eQTL in the result section “Genetic determinants of FHR-4 levels in plasma”. In fact, the *cis*-eQTLs identified are not used in subsequent analysis. Can this part be removed?

Establishing FHR-4 QTLs from multiple datasets and genome-wide analysis methodologies helped us ensure that our findings were not caused by incomplete variant discovery. It was important to look specifically into *cis*-eQTLs as not all *cis*-eQTLs are pQTLs.

The additional value of the *cis*-eQTL analysis is that it allowed us to identify IAMDGC 1.1 (rs1410996) as a possible modifier of FHR-4 levels in plasma. This variant was not found to be an independent pQTL by the two large plasma genome-proteome-wide association studies that we harnessed; however, we did identify it as a possible *cis*-eQTL. Previous studies (Cipriani et al. (2020)⁴⁴, Cipriani et al. (2021)⁴⁵ and Lorés-Motta et al. (2021)⁴³) also found an association between FHR-4 levels and IAMDGC 1.1. These studies used this association to propose that there was an overlap between genetic determinants of AMD susceptibility and genetic modulators of FHR-4 levels in plasma. From this, they concluded that variation in FHR-4 influenced AMD susceptibility. Our refinement of the association between IAMDGC 1.1 and FHR-4 levels revealed that IAMDGC 1.1 did not in fact affect FHR-4 levels, and that its association with FHR-4 variation was entirely attributable to rs7531555.

3. Although the ELISA method used to quantify plasma FHR-4 levels seems to be very specific of FHR-4, the authors should also show the specificity of the antibodies in western blot assays using plasma samples. Particularly in the case of the monoclonal anti-FHR-4 antibody (Cat. #MAB5980, R&D systems), as is the one used in the immunohistochemistry studies, it is important to show that there is no cross-reactivity with other FHR proteins.

To address this key comment, we tested the capture (Cat. #MAB5980, R&D Systems, United States) and detector (Cat. #AF5980, R&D Systems, United States) anti-FHR-4 antibodies on recombinant complement factor H family of proteins and normal human serum by Western blot. The blots were added to the revised manuscript as Supplementary Figure 11.

Robust detection of both FHR-4A and FHR4-B was observed. Both antibodies displayed cross-reactivity with recombinant FHR-3. We did not observe a band consistent with

FHR-3 in normal serum. No cross-reactivity with any of the other recombinants was observed. While cross-reactivity with recombinant FHR-3 on a Western blot does not automatically mean that the antibody will cross-react with the protein in fixed tissue samples, we cannot exclude that part of the positive immunostaining in tissue sections may be attributable to FHR-3.

Our independent IHC analysis has added value as in addition to describing the localization of FHR-4, it reports on potential cross-reactivity with FHR-3. The only previous study of localization of FHR-4 in immunohistochemistry was performed by Cipriani et al (2020), who did not test for cross-reactivity with other complement factor H related proteins (they only tested it for FH and FHL-1). We modified the main manuscript to reflect these new findings. We replaced “confirmed” by “indicated” in the sentence below:

Immunohistochemistry of human donor eyes indicated that FHR-4 was present in Bruch’s membrane and between pillars of the choriocapillaris (see Fig. 2d).

We added the following sentences to the Methods section (“FHR-4 Immunohistochemistry”):

The specificity of the anti-FHR-4 antibody and cross reactivity to complement factor H family of proteins was tested by Western blot. The antibody displayed a cross-reactivity with recombinant FHR-3 (see Supplementary Figure 11). A band consistent with FHR-3 in normal serum was not detected. No cross-reactivity with any of the other recombinants was observed.

We included a discussion on the specific of the antibody used for immunohistochemistry in the Discussions, *i.e.*

We demonstrated that FHR-4 protein was present at the RPE/Bruch’s membrane/choroid interface using both immunohistochemistry and ELISA and that its levels varied with QTLs for liver CFHR4 expression and protein. The antibody that we used for immunohistochemistry displayed cross-reactivity with recombinant FHR-3 on Western blot (see Supplementary Figure 11). While cross-reactivity on a Western blot does not necessarily signal cross-reaction with the protein in fixed tissue samples, we cannot exclude that part of the positive immunostaining in tissue sections may be attributable to FHR-3. The presence of FHR-4 in Bruch’s membrane, choroid and drusen from human donor eyes had been previously demonstrated using immunohistochemistry⁴⁴; however, cross-reactivity with complement factor H related proteins including FHR-3 had not been tested.

Regardless of the results from the Western blot, the capture/detector pairing provides increased specificity and selectivity to target total FHR-4 as demonstrated in Supplementary Figure 5.

4. The sentence “Immunohistochemistry of human donor eyes confirmed that FHR-4 was present in Bruch’s membrane and between pillars of the choriocapillaris (see Fig. 2d).” at the end of the results section “Confirmation of QTL effects in plasma samples”, does not make any sense in that part of the manuscript. Perhaps, it can be relocated in section “Liver QTLs influence FHR-4 levels in RPE/Bruch’s membrane/choroid and vitreous samples”.

The sentence has been moved to the “Genetic Modulators of FHR-4 Influence Levels in RPE/Bruch’s membrane/choroid and Vitreous Samples” section of the manuscript.

5. It has been previously described that FHR-4 plasma levels are elevated in AMD patients compared to controls (Cipriani et al. Nature Communications, 2020, 11:778, <https://doi.org/10.1038/s41467-020-14499-3>). Why do you think you have discrepant results?

Discrepancy between our results and those of Cipriani et al (2020)⁴⁴ can be traced to three main points.

1. Cipriani et al (2020) did not seek to identify causal *CFHR4* QTLs. Cipriani et al (2020) identified *CFHR4* pQTLs by performing a GWAS on FHR-4 levels using 522 controls. They identified a total 543 variants with genome-wide significant association with FHR-4 levels in plasma. The variants that they chose from this list for further analyses were not selected based on refinement approaches designed to identify causal *CFHR4* pQTLs. They simply selected variants that had also been associated with AMD by the IAMDGC GWAS. We took a different approach in our study. We first sought to establish causal QTLs for FHR-4 independently of their association with AMD. We then dissected their association with AMD susceptibility. Cipriani et al (2020) selected IAMDGC 1.1 (tagged by rs1410996 in our study) in their list of variants significantly associated with FHR-4 levels but did not seek to demonstrate that this variant was an independent QTL for *CFHR4*. In our study, we were able to confirm that IAMDGC 1.1 was a QTL for *CFHR4*. We then implemented refinement approaches based on conditional regression analyses, haplotype analyses and association analyses among groups with specific genetic combinations to show that the association between IAMDGC 1.1 and FHR-4 variation reported by Cipriani et al (2020) was in fact entirely attributable to rs7531555.

2. Cipriani et al (2020) did not seek to identify the most likely causal AMD variants. To demonstrate an overlap between AMD-associated variants and genetically driven variations in FHR-4, it is imperative to use AMD variants that are causal, or that are most likely causal variants. If the variants used are not causal, then any detected overlap may be due to linkage disequilibrium, making it potentially spurious. This is particularly relevant to 1q32 locus as it displays a high degree of linkage disequilibrium. In their study, Cipriani et al (2020) simply used the index SNPs for 7 out of the 8 credible sets of variants independently associated with AMD by the IAMDGC GWAS in addition to a SNP tagging the *CFHR3/1* deletion. In our study, we sought to identify the most likely causal AMD variants through conditional genome-wide association analyses, haplotype analyses and diplotype analyses which relied on two large independent cohorts, including the one used by the IAMDGC for their GWAS. We showed that rs61818925 (IAMDGC 1.6) did not differentiate AMD susceptibility and was therefore not causal for AMD. We also demonstrated that IAMDGC 1.1 was not a causal AMD variant either (this is detailed in Supplementary Figure 1, 3 and 4 in addition to the first paragraph of the Results section). Its association with AMD stems from the fact that its minor allele overlaps with the protection-associated A allele at rs800292 (Prot-I62) and the *CFHR3/1* deletion haplotype. Having demonstrated this, we were able to robustly investigate possible overlaps between independent *CFHR4* QTLs and causal AMD-associated variants.

3. Cipriani et al (2020) did not disentangle AMD variants from *CFHR4* QTLs. Rather, they used linkage disequilibrium to imply that variants responsible for variations in FHR-4 levels also affected AMD susceptibility without performing the necessary analyses to prove causality. Our manuscript sought to first establish causal AMD variants, then to identify independent QTLs for FHR-4 levels. We were then able to dissect associations between causal AMD variants with variation in FHR-4, and between *CFHR4* QTLs and AMD susceptibility. In this way, we were able to overcome the challenge of LD and show that the variants affecting FHR-4 levels are not the variants that confer AMD risk.

Differences in FHR-4 levels between individuals with late AMD and controls has only been established in the combined Cambridge and EUGENDA cohorts used by Cipriani et al (2020) and in later publications by Cipriani et al (2021)⁴⁵ and Lores-Motta et al (2021)⁴³. Our study is the first to assess this difference in an independent cohort. We could not replicate this finding. Since we do not have access to the raw data from Cipriani et al (2020), we can only make educated conjectures on why they observed that patients with late AMD have higher FHR-4 levels as compared to healthy controls. Because they have an opposite differential effect on FHR-4 levels and are present on AMD risk haplotypes, the frequencies of the minor alleles at rs61818956 and rs10494745 are likely to strongly influence the variability in FHR-4 levels among groups of individuals carrying high-risk haplotypes at the Chr1 AMD locus, which are more common among subjects with AMD. The presence of the level-reducing T at rs7531555 on protective haplotypes, which are more common among controls, is also likely to influence observed differences in levels between cases and controls.

It should however be noted that our study demonstrates that previously reported differences in FHR-4 levels between controls and patients with AMD is in fact inconsequential. Our rigorous analyses prove that genetic variation in FHR-4 levels is completely independent from AMD susceptibility. Many factors may explain why FHR-4 levels were observed to be higher among AMD patients; however, none of them influence AMD susceptibility in a significant way.

Some of this discussion was already present in our original submission. We added a sentence to the third paragraph of the Discussions (underlined below) to explicitly discuss reports of higher FHR-4 levels in AMD patients:

Associations with FHR-4 levels reported with the AMD-associated rs1410996 (IAMDC 1.1) and rs1061170 (CFH Y402H, IAMDC 1.2)^{43-45,60} are attributable to rs7531555 and to the combination of rs61818956 and rs10494745, respectively. Because they have an opposite differential effect on FHR-4 levels and are present on AMD risk haplotypes, the frequencies of the minor allele at rs61818956 and rs10494745 are likely to strongly influence the variability in FHR-4 levels among groups of individuals carrying high-risk haplotypes at the Chr1 AMD locus and among patients with AMD⁴³⁻⁴⁵. Significantly lower FHR-4 that we and others⁴³⁻⁴⁵ observed among individuals carrying the protective A allele at rs800292, which is more frequent among controls, as compared to subjects carrying the risk-associated C allele at rs1061170, are likely to be driven by the level-reducing T allele at rs7531555. Altogether, the presence of opposing QTLs on AMD risk haplotypes, which are more common among AMD patients, combined with the presence of the level-reducing T at rs7531555 on protective haplotypes, which are more common among controls, may explain previous findings, established using small sample sizes, that FHR-4 levels were elevated among individuals with AMD⁴⁴.

Regarding the inadequacy of IAMDC 1.1 to assess overlaps between AMD-associated variants and genetic modulators of FHR-4 levels, we stated in the Discussions

Causality for protection at the Chr1 AMD locus is therefore more likely to originate from rs800292 or the deletion of CFHR3/1 than from rs1410996. This observation makes the use of IAMDC 1.1 to demonstrate overlaps between specific phenotypes, such as the concentration of various proteins in serum and tissue, and AMD-associated variants^{43-45,60}, problematic. Previous work has shown that IAMDC 1.1 influences the level of at least 40 serum proteins, with only six of them associated with AMD⁶⁰. It is therefore essential to carefully refine both protein-specific and AMD associations with this variant. In our study, we were for instance able to show that associations between plasma FHR-4 levels and

rs1410996 were in fact entirely attributable to rs7531555, a robust CFHR4 QTL with no independent association with AMD.

In addition to these changes, differences in cohorts studied and other aspects of study design between our study and that of Cipriani et al. (2020) are detailed in Supplementary Table 8 of the revised manuscript. Supplementary Table 7 of the revised manuscript compares findings by Cipriani et al. (2020) and our findings side by side. The table shows that several of our findings explain those of Cipriani et al. (2020). Finally, we added an extended discussion of similarities and discrepancies of our findings with those reported by Cipriani et al (2020) in Supplementary Text 1 of the revised manuscript.

6. Why should “liver” be included in the titles “Liver QTLs influence FHR-4 levels in RPE/Bruch’s membrane...” and “Liver QTLs are not independently associated with AMD”? I understand that FHR-4 is mainly produced by the liver, but are the authors excluding the effect of these QTLs in any other cell type that may express the protein?

No significant amount of CFHR4 mRNA has been detected in cell types other than liver cells. All FHR-4 protein available systemically is therefore produced in the liver. Since CFHR4 is not expressed in ocular tissue, all FHR-4 protein present in ocular compartment originates in the liver. The use of “liver” in the title of several results section is therefore superfluous and has been removed from the revised manuscript.

7. The section of the results “Liver QTLs are not independently associated with AMD” seems redundant to me, as this part could be deduced from the first section of the results, where the 3 SNPs that determine susceptibility to the disease are described. Also, I am not sure what the rationale is for result section “Variation in FHR-4 plasma levels does not influence AMD risk or protection”, since the authors described that FHR-4 plasma levels are not significantly different between controls, early/intermediate AMD, and late AMD samples (Suppl. Figure S5). Therefore, I suggest deleting some of the sections or, at least, simplifying them.

The aim of the section entitled “Liver QTL are not independently associated with AMD” was to explain why all three independent CFHR4 QTLs are significantly associated with AMD. It is not sufficient to rely on our analysis demonstrating that discriminants of AMD susceptibility can be reduced to the combination rs1061170, rs800292 and a SNP tagging the CFHR3/1 deletion to explain these associations. To do so in a robust way, we showed that when we account for the effect of most likely AMD causal variants, associations between CFHR4 QTLs and AMD are lost.

To clarify this, we changed the title of the section to “Associations of CFHR4 QTLs with AMD are Attributable to rs1061170, rs800292 and the CFHR3/1 Deletion Haplotype”. We also modified the paragraph to explicitly state that our goal was to explain the association between CFHR4 QTLs and AMD, i.e.

Independent CFHR4 QTLs influence the concentration of FHR-4 in plasma and at the primary location of AMD-associated pathology. Single-variant association analyses performed using the Utah & Iowa cohort and validated against the IAMDGC cohort indicated that the minor allele at rs10494745 (A) and rs61818956 (T), which have opposing effects on FHR-4 levels, were both associated with an increased risk for AMD (OR_{IAMDGC} : 1.68 [1.60; 1.77], $p = 7.8e-88$ and OR_{IAMDGC} : 1.52 [1.45; 1.58], $p = 7.0e-82$, respectively), while the FHR-4 lowering allele at rs7531555 (T) is associated with protection against AMD (OR_{IAMDGC} : 0.58 [0.56; 0.61], $p = 2.7e-135$); see Fig. 2a and Supplementary Tables 3 and 4. To explain these associations, we performed independent haplotype analyses on AMD status among the Utah & Iowa and IAMDGC cohorts, see Fig. 3a.

The aim of the section “*Variation in FHR-4 Plasma Levels does not Influence AMD Risk or Protection*” was to demonstrate the absence of association between variation in FHR-4 and AMD susceptibility using two large, independent cohorts. Stratifying the Utah & Iowa and IAMDGC cohorts based on both AMD susceptibility and FHR-4 levels allowed us to validate the absence of association between AMD and FHR-4 variations in the most robust way. We do agree that in the original manuscript this section appears to overlap with results from other sections. To improve clarity and readability, we simplified it, shortened it and, since it relies on a similar stratification approach, merged it with the section originally entitled “*FHR-4 Plasma and Ocular Levels do Not Influence AMD Disease Progression*”, which we renamed “*FHR-4 Plasma and Ocular Levels do Not Influence AMD Susceptibility or Disease Progression*”. The paragraph now reads:

We examined the differential effect of FHR-4 variation on AMD susceptibility in the two large, independent Utah & Iowa and IAMDGC case/control cohorts. To do so, we assessed associations of haplotypes based on the AMD-associated variants rs1061170, rs800292 and rs12144939 and CFHR4 QTLs (associated with reduced, baseline or elevated FHR-4 levels) with AMD status (see Fig. 3a). AMD odds ratios in the Prot-I62 and Neutral haplotype groups did not differ significantly between subjects with reduced FHR-4 levels as compared to those with baseline concentrations (see Fig. 3c and Supplementary Figure 10). Among chromosomes with a neutral haplotype, no significant effect on AMD odds ratios were detected for elevated FHR-4 levels (Supplementary Figure 10). Among chromosomes with a risk haplotype, both reduced (A at rs10494745) and elevated (T at rs61818956) FHR-4 levels are associated with a 1.1-fold increase in AMD odds ratios as compared to baseline ($p = 0.002$ for both variants; see Fig. 3d). Altogether, genetic variation in FHR-4 levels do not influence the protection against AMD conferred by the Prot-I62 haplotype, the risk for AMD associated with the Risk haplotype or the lack of protection or risk associated with the Neutral haplotype, see Fig. 3e.

Minor comments

8. To mention the Figures and Tables in order of appearance in the manuscript. For instance, figure 1B should be fig.1A and vice versa, as Fig. 1B is mentioned before (introduction). The same situation applies to supplemental tables and figures. - Figure 2A. I suggest including the schematic of the genes where the SNPs are located at the bottom of the graph (similar to fig. 1A), as it will serve as a guide for locating the variants. - Suppl. Figure 10C. To change the colour of the black circles to depict samples with 2 copies in the graph, so the SD bars can be seen.

We reordered all figures, tables, supplementary figures, supplementary figures and supplementary text in order of appearance in the main text. The exact location of the *CFH*, *CFHR1*, *CFHR2*, *CFHR3*, *CFHR4* and *CFHR4* has been added to Figure 1b and Figure 2a. The color of the black circles was changed in Supplementary Figure S10 (Supplementary Figure 5 in the revised manuscript), as shown below.

Reviewer 2

The 3-page introduction is a review of AMD genetics and references are mostly the authors' papers. This could be shortened considerably and could include references from other groups. A paper relevant to the topic was not mentioned or addressed. Raychaudhuri S, Ripke S, Li M, Neale BM, et al. Associations of CFHR1-CFHR3 deletion and a CFH SNP to age-related macular degeneration are not independent. Nat Gen 2010; 42:553-555.

Our manuscript overlaps with several areas of AMD research, including genetic associations within the *CFH-CFHR5* locus and the differential contribution of FHR proteins including FHR-4 to AMD susceptibility. Laying down a comprehensive description of current knowledge in these areas is challenging because of discrepancies and inconsistencies between past studies. We have done our best to make the Introduction as concise as possible without compromising on descriptions of the current state of the field. Currently the Introduction contains only information that is key to understanding the remainder of the manuscript.

We have added the reference mentioned by the Reviewer to the revised manuscript. As it stands, less than 20% of references in the Introduction are from our group or co-authors.

Figure 4 - small numbers. What was the effect of all other genetic variants given that this group has a specific risk variant among many others that could also influence risk. Several cells in this figure have fewer than 5 individuals. What were results without making these subgroups?

Large and well-curated longitudinal clinical datasets that also contain genetic information for each participant are uncommon. This makes our analysis of association between FHR-4 variation and AMD progression incredibly valuable.

Including all other genetic variants in our analysis would increase variability considerably, mainly because of factors that are independent from risk at the Chr1 AMD locus or FHR-4 variations. Our analysis was specifically designed to reduce the amount of variability associated with both AMD and FHR-4 variation. Considering individuals homozygous for risk alleles at rs1061170 and with no risk alleles at the Chr10 AMD locus ensured that cases of AMD were predominantly caused by Chr1-driven complement dysregulation. Restricting our analyses to subjects with homozygosity at rs10494745 (reduced FHR-4) and rs61818956 (elevated FHR-4) ensures that we are comparing groups of individuals with the largest difference in FHR-4 levels. Our approach therefore minimizes the risk for spurious findings caused by small sample sizes and ensures that we are assessing the true contribution of FHR-4 variation to disease progression. These points are emphasized in the revised manuscript with the following sentences (section “*FHR-4 Plasma and Ocular Levels do Not Influence AMD Susceptibility or Disease Progression*”):

Therefore, we assessed if variations in FHR-4 levels impacted the course of AMD among patients with disease driven primarily by AP dysregulation. To do so, we analyzed longitudinal clinical data collected from 317 patients with a Risk/Risk diplotype and no risk alleles at the Chr10 AMD locus¹⁶. To ensure that comparisons were performed between groups of individuals with the largest difference in FHR-4 levels, we selected patients on the basis on homozygosity at rs61818956 (TT, FHR-4[↑] group) and rs10494745 (AA, FHR-4[↓] group), with the reference group designated FHR-4[↔] (CC at rs61818956, GG at rs10494745).

Age at first visit and AMD status at first visit needed to be accounted for in our survival analysis to ensure that this analysis was not biased by when (at which disease stage)

patients were recruited in. Not including this variable in Cox-PH regression models would therefore yield inaccurate predictions. The need for this approach has been proved in clinical publications and is widely used in clinical studies (see for instance Schmitz-Valckenberg et al 2021, <https://jamanetwork.com/journals/jamaophthalmology/fullarticle/2788507>).

Much effort was made to determine which CFH snp among a few in high LD were related to AMD, discounting one of them based on results in 0.6 to 0.7% of the cases and controls. Whereas the main goal was to show that previously published CFHR-4 results were not accurate.

Previous studies, including Cipriani et al (2020)⁴⁴, argued that FHR-4 levels influenced AMD susceptibility by demonstrating an overlap between certain genetic determinants of AMD susceptibility and variation in FHR-4 levels in plasma. To demonstrate an overlap between AMD-associated variants and genetically driven variations in FHR-4, it is imperative to use AMD variants that are causal, or that are most likely causal variants. If the variants used are not causal, then any detected overlap may be due to linkage disequilibrium, making it potentially spurious. This is particularly relevant to 1q32 locus as it displays a high degree of linkage disequilibrium. Cipriani et al (2020) simply used the index SNPs for 7 out of the 8 credible sets of variants independently associated with AMD by the IAMDGC GWAS in addition to a SNP tagging the *CFHR3/1* deletion. In our study, we showed that rs61818925 (IAMDGC 1.6) did not differentiate AMD susceptibility and was therefore not causal for AMD. We also demonstrated that IAMDGC 1.1 was not a causal AMD variant either (this is detailed in Supplementary Figure 1, 2, 3 and 4 in addition to the first paragraph of the Results section). Its association with AMD stems from the fact that its minor allele overlaps with the protection-associated A allele at rs800292 and the *CFHR3/1* deletion haplotype. Having demonstrated this, we were able to robustly investigate possible overlaps between independently identified *CFHR4* QTLs and causal AMD-associated variants.

Refinement of associations with AMD within the Chr1 AMD locus is therefore a key part of our manuscript.

Results require independent confirmation.

Our findings were cross-validated by using two independent cohorts, the Utah & Iowa case/control cohort and IAMDGC case/control cohort. The IAMDGC cohort was the one used for the largest AMD GWAS to date and represents a reference in the field. Identification of QTLs relied on the two largest plasma genome-proteome-wide association studies to date and the Genotype-Tissue Expression Project. The effect of the three independent *CFHR4* QTLs on FHR-4 levels in plasma explain findings from two previous publications, as shown in Supplementary Table 7 of the revised manuscript. We were not able to access an independent sufficiently well-curated longitudinal clinical dataset that would have allowed for the validation of our analysis of association between FHR-4 variation and AMD progression. We did however provide all necessary information for this analysis to be replicated in future independent studies.

Reviewer 3

It was a tremendous pleasure reading this manuscript by Dr Zouache and co-authors reporting that variation in FHR4 levels in the human body are not significantly associated with susceptibility to AMD as well as disease progression. The analysis here appears to be vigorously, critically, and transparently done. Here, the Authors arrived at the conclusion that variation in FHR-4 is unlikely to be related to AMD risk because both elevated and reduced plasma and ocular tissue levels of FHR-4 coincide with a higher risk for AMD. I have several very humble suggestions for the Authors:

1. The key conclusions of this manuscript is not consistent with that by Cipriani et al., 2020 (Reference #43, main text). There is thus a robust, collegial and constructive scientific discussion to be had that will strongly benefit readers and the AMD research community. The key point up for discussion would be the findings in this submission that "Variation in FHR-4 Plasma Levels does Not Influence AMD Risk or Protection" whereas Cipriani et al., observed that "Systemic FHR-4 levels are elevated in advanced AMD cases". I would encourage the Authors here to discuss their present findings side by side with Cipriani et al; as one key observable difference was that whereas Cipriani et al., evaluated 484 late AMD patients and 522 phenotyped controls, the present submission evaluates samples in 486 individuals from the Utah & Iowa cohort (median age 75.0 IQR 13.0) with no AMD (235 subjects), early/intermediate AMD (120 subjects) or late AMD (131 subjects) (lines 248 to 250, main text).

Discrepancy between our results and those of Cipriani et al (2020)⁴⁴ can be traced to three main points.

1. Cipriani et al (2020) did not seek to identify causal *CFHR4* QTLs. Cipriani et al (2020) identified *CFHR4* pQTLs by performing a GWAS on FHR-4 levels using 522 controls. They identified a total 543 variants with genome-wide significant association with FHR-4 levels in plasma. The variants that they chose from this list for further analyses were not selected based on refinement approaches designed to identify causal *CFHR4* pQTLs. They simply selected variants that had also been associated with AMD by the IAMDGC GWAS. We took a different approach in our study. We first sought to establish causal QTLs for FHR-4 independently of their association with AMD. We then dissected their association with AMD susceptibility. Cipriani et al (2020) selected IAMDGC 1.1 (tagged by rs1410996 in our study) in their list of variants significantly associated with FHR-4 levels but did not seek to demonstrate that this variant was an independent QTL for *CFHR4*. In our study, we were able to confirm that IAMDGC 1.1 was a QTL for *CFHR4*. We then implemented refinement approaches based on conditional regression analyses, haplotype analyses and association analyses among groups with specific genetic combinations to show that the association between IAMDGC 1.1 and FHR-4 variation reported by Cipriani et al (2020) was in fact entirely attributable to rs7531555.

2. Cipriani et al (2020) did not seek to identify the most likely causal AMD variants. To demonstrate an overlap between AMD-associated variants and genetically driven variations in FHR-4, it is imperative to use AMD variants that are causal, or that are most likely causal variants. If the variants used are not causal, then any detected overlap may be due to linkage disequilibrium, making it potentially spurious. This is particularly relevant to 1q32 locus as it displays a high degree of linkage disequilibrium. In their study, Cipriani et al (2020) simply used the index SNPs for 7 out of the 8 credible sets of variants independently associated with AMD by the IAMDGC GWAS in addition to a SNP tagging the *CFHR3/1* deletion. In our study, we sought to identify the most likely causal AMD variants through conditional genome-wide association analyses, haplotype analyses and diplotype analyses which relied on two large independent cohorts, including the one used by the IAMDGC for their GWAS. We showed that rs61818925 (IAMDGC 1.6) did not differentiate AMD susceptibility and was therefore not causal for AMD. We also

demonstrated that IAMDGC 1.1 was not a causal AMD variant either (this is detailed in Supplementary Figure 1, 3 and 4 in addition to the first paragraph of the Results section). Its association with AMD stems from the fact that its minor allele overlaps with the protection-associated A allele at rs800292 (Prot-I62) and the *CFHR3/1* deletion haplotype. Having demonstrated this, we were able to robustly investigate possible overlaps between independent *CFHR4* QTLs and causal AMD-associated variants.

3. **Cipriani et al (2020) did not disentangle AMD variants from *CFHR4* QTLs.** Rather, they used linkage disequilibrium to imply that variants responsible for variations in FHR-4 levels also affected AMD susceptibility without performing the necessary analyses to prove causality. Our manuscript sought to first establish causal AMD variants, then to identify independent QTLs for FHR-4 levels. We were then able to dissect associations between causal AMD variants with variation in FHR-4, and between *CFHR4* QTLs and AMD susceptibility. In this way, we were able to overcome the challenge of LD and show that the variants affecting FHR-4 levels are not the variants that confer AMD risk.

Differences in FHR-4 levels between individuals with late AMD and controls has only been established in the combined Cambridge and EUGENDA cohorts used by Cipriani et al (2020) and in later publications by Cipriani et al (2021)⁴⁵ and Lores-Motta et al (2021)⁴³. Our study is the first to assess this difference in an independent cohort. We could not replicate this finding. Since we do not have access to the raw data from Cipriani et al (2020), we can only make educated conjectures on why they observed that patients with late AMD have higher FHR-4 levels as compared to healthy controls. Because they have an opposite differential effect on FHR-4 levels and are present on AMD risk haplotypes, the frequencies of the minor alleles at rs61818956 and rs10494745 are likely to strongly influence the variability in FHR-4 levels among groups of individuals carrying high-risk haplotypes at the Chr1 AMD locus, which are more common among subjects with AMD. The presence of the level-reducing T at rs7531555 on protective haplotypes, which are more common among controls, is also likely to influence observed differences in levels between cases and controls.

It should however be noted that our study demonstrates that previously reported differences in FHR-4 levels between controls and patients with AMD is in fact inconsequential. Our rigorous analyses proves that genetic variations in FHR-4 levels are completely independent from AMD susceptibility. Many factors may explain why FHR-4 levels were observed to be higher among AMD patients; however, none of them influence AMD susceptibility in a significant way.

Some of this discussion was already present in our original submission. We added a sentence to the third paragraph of the Discussions (underlined below) to explicitly discuss reports of higher FHR-4 levels in AMD patients:

Associations with FHR-4 levels reported with the AMD-associated rs1410996 (IAMDGC 1.1) and rs1061170 (CFH Y402H, IAMDGC 1.2)^{43-45,60} are attributable to rs7531555 and to the combination of rs61818956 and rs10494745, respectively. Because they have an opposite differential effect on FHR-4 levels and are present on AMD risk haplotypes, the frequencies of the minor allele at rs61818956 and rs10494745 are likely to strongly influence the variability in FHR-4 levels among groups of individuals carrying high-risk haplotypes at the Chr1 AMD locus and among patients with AMD⁴³⁻⁴⁵. Significantly lower FHR-4 that we and others⁴³⁻⁴⁵ observed among individuals carrying the protective A allele at rs800292, which is more frequent among controls, as compared to subjects carrying the risk-associated C allele at rs1061170, are likely to be driven by the level-reducing T allele at rs7531555. Altogether, the presence of opposing QTLs on AMD risk haplotypes, which are more common among

AMD patients, combined with the presence of the level-reducing T at rs7531555 on protective haplotypes, which are more common among controls, may explain previous findings, established using small sample sizes, that FHR-4 levels were elevated among individuals with AMD⁴⁴.

Regarding the inadequacy of IAMDGC 1.1 to assess overlaps between AMD-associated variants and genetic modulators of FHR-4 levels, we stated in the Discussions

Causality for protection at the Chr1 AMD locus is therefore more likely to originate from rs800292 or the deletion of CFHR3/1 than from rs1410996. This observation makes the use of IAMDGC 1.1 to demonstrate overlaps between specific phenotypes, such as the concentration of various proteins in serum and tissue, and AMD-associated variants^{43–45,60}, problematic. Previous work has shown that IAMDGC 1.1 influences the level of at least 40 serum proteins, with only six of them associated with AMD⁶⁰. It is therefore essential to carefully refine both protein-specific and AMD associations with this variant. In our study, we were for instance able to show that associations between plasma FHR-4 levels and rs1410996 were in fact entirely attributable to rs7531555, a robust CFHR4 QTL with no independent association with AMD.

Of note, two GWAS conducted among 252 controls from the Cambridge cohort (Cipriani et al. 2021, <https://doi.org/10.1016/j.ajhg.2021.05.015>)⁴⁵ and 202 controls from the EUGENDA cohort (Lorés-Motta et al. 2021, <https://doi.org/10.1016/j.ajhg.2021.06.002>)⁴³, which were combined in the original publication from Cipriani et al. (2020), reported findings that are consistent with ours (this analysis was added as Supplementary Table 7). The two studies relied either on objective variant prioritization technique (variants were selected independently from their potential involvement in AMD) or on conditional analyses for refinements. Lorés-Motta et al. (2021) identified rs10494745 (not prioritized by Cipriani et al. 2020) and a perfect proxy for rs1410996 (IAMDGC 1.1) as FHR-4 pQTLs (whose effect we show is attributable to rs7531555). The two variants reported by Cipriani et al. (2021) were rs4085749, which is a perfect proxy for rs7531555 ($r^2 = 1.0$ and $D' = 1.0$), and rs12047098, which is in strong LD with rs7531555 ($r^2 = 0.826$ and $D' = 0.949$). None of these variants were prioritized in the original publication from Cipriani et al. (2020). The fact that neither study identified all independent *CFHR4* QTLs supports the idea of a bias due to small sample size in the three studies and demonstrates the importance of adequate variant prioritization techniques and refinements. It also shows the importance of using multiple datasets and genome-wide analysis methodologies for complete QTL discovery.

Since we do not have access to the raw data from Cipriani et al. (2020) and given that our methodology differs in several ways we feel that including this detailed discussion in the Results section would be inadequate. We have however included most of this discussion in an additional Supplementary Material section (Supplementary Text 1) entitled “Extended discussion on similarities and discrepancies of findings with those reported by Cipriani et al. (2020)⁴⁴, Cipriani et al. (2021)⁴⁵ and Lorés-Motta et al. (2021)⁴³”. In addition, as suggested by the reviewer, we included a table describing in detail how our study differs from Cipriani et al. (2020)⁴⁴, Cipriani et al. (2021)⁴⁵ and Lorés-Motta et al. (2021)⁴³ in Supplementary Table 8.

2. The Authors observed that single-variant association analyses performed using the Utah & Iowa cohort and validated against the IAMDGC cohort indicated that rs10494745 and rs61818956, which have opposing effects on FHR-4 levels, were both associated with an increased risk for AMD. Could they explain whether the opposing effect of both key variants on FHR-4 levels were solely responsible for them not observing an association

between FHR-4 levels and disease risk? This is a key question that should be clarified for the reader because Cipriani et al., also presented their analysis based on IAMDC SNPs on the chromosome 1 locus. Perhaps, a vigorous analysis on ‘where they differ between the 2 papers’ would be helpful, as both studies enrolled participants of predominantly European ancestry, and the genetic architecture, haplotypic variation and haplotypic relationships should not differ by much between the 2 studies.

The fact that rs10494745 and rs61818956 have opposite effects on FHR-4 variation is not responsible for the absence of association between FHR-4 levels and disease risk. This absence of association stems from the fact that these two QTLs for *CFHRA* are not independently associated with AMD.

We showed through single-variant association analyses that rs10494745 and rs61818956 are associated with an increased risk for developing AMD. We then proceeded to refine these associations by using conditional regression analyses and haplotype analyses. Haplotype analyses showed that the FHR-4 reducing allele at rs10494745 (A) resides almost exclusively on a haplotype carrying the risk-associated C allele at rs1061170 ($D' = 0.96$ and $R^2 = 0.21$). In addition, we observed that association analyses of rs10494745 with AMD status lost their significance when conditioned on rs1061170. These two results indicate that rs10494745 does not influence disease risk. The association identified through single-variant analysis is in fact entirely attributable to the rs10494745 allele reducing FHR-4 levels coinciding with the risk-associated C allele at rs1061170. Similarly, the FHR-4 increasing allele at rs61818956 (T) exists primarily as part of a risk-associated haplotype (H5, 10.1% of 1000 G EUR chromosomes and 11.2% and 12.3% of chromosomes from Utah & Iowa and IAMDC controls, respectively) that also carries the C allele at rs1061170. In addition, we observed that association analyses of rs61818956 with AMD status lost their significance when conditioned on rs1061170. The association identified through single-variant analysis is therefore again entirely attributable to the rs61818956 allele increasing FHR-4 levels coinciding with the risk-associated C allele at rs1061170.

Reciprocally, we showed that AMD causal variants were not independently associated with FHR-4 variation.

These two sets of analyses showed that there is no overlap between AMD-associated variants and haplotypes and the genetic determinants of FHR-4 levels in plasma and ocular tissue. It is this independence that allowed us to show that FHR-4 levels did not influence disease risk.

As noted by the Reviewer, the difference in conclusions between our study and that of Cipriani is unlikely to be caused by patient selection or variation in genetic architecture or haplotypes between cohorts. Differences between our studies and plausible reasons for discrepancies in final conclusions are detailed in our response to the previous comment.

3. In its present form, the manuscript is written with many rs-ids and terminologies (for example H7, line 313 main text; H5 line 310) describing the extremely complicated relationship between FHR-4 levels, genetic variants (and haplotypes) associated with AMD risk, as well as genetic variants (and haplotypes) associated with FHR-4 levels (QTLs). Even for a self-confessed AMD-enthusiast such as myself, I had to struggle through the rs-ids between text and figures multiple times to reconcile them. Could the Authors consider simplifying the presentation to make it easier for the reader to appreciate the question and the science behind it?

We did our best to strike a balance between current writing conventions (which for instance prevent us from using notations used for protein alterations to refer to genetic

variants) and readability. To improve readability, we took the following steps in the revised manuscript:

1. We specified the allele and associated effect on AMD susceptibility or FHR-4 levels every time we referred to a variant,
2. We added alternative labels, major and minor alleles, and direction of effect for AMD association for each variant on Fig. 1b,
3. We added alternative labels, major and minor alleles, and direction of effect for FHR-4 levels for each variant on Fig. 2a,
4. We removed the labels H1, H2, H3, H4 for haplotypes shown in Fig. 1c. These labels are now exclusively used for Fig. 3a,
5. Whenever possible we specified the relevant alleles carried by haplotypes that were discussed and their associated effect.

Reviewers' Comments:

Reviewer #1:

Remarks to the Author:

The authors have satisfactorily addressed the suggestions and comments made during the review process. I believe that the changes introduced in the new version have significantly improved the manuscript. I have only one doubt and one suggestion regarding my comment 3: Were the same amount of all recombinant proteins used for Western blots with the anti-FHR4 antibodies? If this is the case, then I think the cross-reactivity of the monoclonal antibody with FHR-3 can be considered negligible, and as such I would mention this in the manuscript.

I have no further comments.

Reviewer #3:

Remarks to the Author:

The Authors have been extremely responsive to the comments raised and discussed in the first round of review. Specifically, they should be celebrated for their very transparent and thorough analysis (systematic fine mapping of causal variants that was not as thoroughly done by Cipriani et al.,) and their candid discussion.

Usually though, linkage disequilibrium is a good thing, as it tags functional variants and is the basis for successful GWAS designs (SNPS located within the same block will replicate if the signal is a true positive).

Here, the Authors go over and beyond to show that when very complex genetic situations arise (such as this one, whereby genetics, expression, and proteomics are considered together), linkage disequilibrium may give rise to SPURIOUS findings if complex relationships between exposures are not thoroughly accounted for. The Authors do a beautiful job in the revision.

Additional comments on responses to Reviewer #2:

The Authors have mostly responded to Reviewer #2 comments and suggestions to the best of their ability.

However, in my very humble view, they did not respond directly to the comment regarding the small numbers in Figure #4.

Reviewer #2 sought clarification from the Authors regarding the very small number of participants in the different AMD severity grades stratified by FHR-4 level (Figure 4A) and conversions (Figure 4B).

Whereas I appreciate the complete transparency of the Authors in presenting the data as it stood, the Authors should consider discussing the impact of the limited numbers in Figures 4A and B with the view to aid the reader as to potential limitations (statistical uncertainty due to small numbers), and thus, the inability to draw firm conclusions based on these numbers alone.

Levels of Complement Factor H-related 4 Protein do Not Influence Susceptibility to Age-related Macular Degeneration or its Course of Progression (Zouache *et al.*) - Response to Reviewer Comments.

Reviewer 1

The authors have satisfactorily addressed the suggestions and comments made during the review process. I believe that the changes introduced in the new version have significantly improved the manuscript. I have only one doubt and one suggestion regarding my comment 3: Were the same amount of all recombinant proteins used for Western blots with the anti-FHR4 antibodies? If this is the case, then I think the cross-reactivity of the monoclonal antibody with FHR-3 can be considered negligible, and as such I would mention this in the manuscript.

I have no further comments.

The same amount of all recombinant proteins was indeed used in the western blot studies. We modified the Discussions section of the manuscript to specify, as suggested by the reviewer, that the cross-reactivity of the monoclonal antibody with FHR-3 can in fact be considered negligible. The modified Discussions now states:

The monoclonal antibody used for immunohistochemistry displayed cross-reactivity with recombinant FHR-3 on Western blot but did not detect this protein in normal serum (see Supplementary Figure 11). Since equimolar concentration of recombinant protein was compared in the assay, the cross-reactivity of the antibody with FHR-3 is negligible.

In addition, the sentence “Equimolar concentration of recombinant protein was used for this assay” was added to the Methods section describing the validation of the antibodies for Western blot.

Additional Comments on Responses to Reviewer 2

The Authors have mostly responded to Reviewer #2 comments and suggestions to the best of their ability. However, in my very humble view, they did not respond directly to the comment regarding the small numbers in Figure #4. Reviewer #2 sought clarification from the Authors regarding the very small number of participants in the different AMD severity grades stratified by FHR-4 level (Figure 4A) and conversions (Figure 4B). Whereas I appreciate the complete transparency of the Authors in presenting the data as it stood, the Authors should consider discussing the impact of the limited numbers in Figures 4A and B with the view to aid the reader as to potential limitations (statistical uncertainty due to small numbers), and thus, the inability to draw firm conclusions based on these numbers alone.

The following paragraph was added to the Discussions:

Examination of longitudinal clinical data suggests that variation in FHR-4 is unlikely to influence the risk of progression to late AMD or slow the progression of the disease. It should be noted that the survival analysis that we performed is limited by small sample size and by heterogeneities in AMD severity at first visit between eyes. To address this and to reduce the amount of variability associated with both AMD and FHR-4 variation, we considered individuals homozygous for risk alleles at rs1061170 and with no risk alleles at the Chr10 AMD locus so that cases of AMD were predominantly caused by Chr1-driven complement dysregulation. Restricting our analyses to subjects with homozygosity at rs10494745 (reduced FHR-4) and rs61818956 (elevated FHR-4) ensured that we compared groups of individuals with the largest difference in FHR-4 levels. Given the small number of eyes that met these criteria, additional AMD-associated variants may have influenced our findings.

Previous work^{62,63} has shown that in addition to the CFH-CFH5 and ARMS2/HTRA1 regions, the C2-CFB and C3 genes may have a significant impact on AMD progression. Our analysis did not allow us to identify or account for the possible effect of confounders associated specifically with progression to neovascular AMD but not cRORA⁶² (or vice-versa). Confirmation of our findings using large and well-curated longitudinal clinical datasets is therefore warranted.